# Efficient Symbolic Policy Learning with Differentiable Symbolic Expression

**Jiaming Guo[1]   Rui Zhang[1]   Shaohui Peng[2]   Qi Yi[1,3,4]   Xing Hu[1,5]**
**Ruizhi Chen[2]   Zidong Du[1,5]   Xishan Zhang[1,4]   Ling Li[2,6]   Qi Guo[1]   Yunji Chen[1,6] ***

[1] SKL of Processors, Institute of Computing Technology, CAS, Beijing, China
[2] Intelligent Software Research Center, Institute of Software, CAS, Beijing, China
[3] University of Science and Technology of China, USTC, Hefei, China
[4] Cambricon Technologies
[5] Shanghai Innovation Center for Processor Technologies, SHIC, Shanghai, China
[6] University of Chinese Academy of Sciences, UCAS, Beijing, China

{guojiaming, zhangrui}@ict.ac.cn, pengshaohui@iscas.ac.cn, yiqi@mail.ustc.edu.cn
huxing@ict.ac.cn, ruizhi@iscas.ac.cn {duzidong,zhangxishan}@ict.ac.cn
liling@iscas.ac.cn, {guoqi,cyj}@ict.ac.cn

## Abstract

Deep reinforcement learning (DRL) has led to a wide range of advances in sequential decision-making tasks. However, the complexity of neural network policies makes it difficult to understand and deploy with limited computational resources. Currently, employing compact symbolic expressions as symbolic policies is a promising strategy to obtain simple and interpretable policies. Previous symbolic policy methods usually involve complex training processes and pre-trained neural network policies, which are inefficient and limit the application of symbolic policies. In this paper, we propose an efficient gradient-based learning method named Efficient Symbolic Policy Learning (ESPL) that learns the symbolic policy from scratch in an end-to-end way. We introduce a symbolic network as the search space and employ a path selector to find the compact symbolic policy. By doing so we represent the policy with a differentiable symbolic expression and train it in an off-policy manner which further improves the efficiency. In addition, in contrast with previous symbolic policies which only work in single-task RL because of complexity, we expand ESPL on meta-RL to generate symbolic policies for unseen tasks. Experimentally, we show that our approach generates symbolic policies with higher performance and greatly improves data efficiency for single-task RL. In meta-RL, we demonstrate that compared with neural network policies the proposed symbolic policy achieves higher performance and efficiency and shows the potential to be interpretable.

## 1   Introduction

With the development of deep neural networks as general-purpose function approximators, deep reinforcement learning (DRL) has achieved impressive results in solving sequential decision-making tasks [1, 2]. In DRL, the policies are commonly implemented as deep neural networks which involve tremendous parameters and thousands of nested non-linear operators. Despite the excellent representation ability, the neural network (NN) is very complex, making it difficult to understand, predict the behavior and deploy with limited computational resources.

---

*Corresponding Author.

37th Conference on Neural Information Processing Systems (NeurIPS 2023).

With the high academic and industrial interest in interpretable and simple RL policy, some works [3, 4, 5] propose to learn the symbolic policy which is a symbolic expression composing variables, constants, and various mathematical operators. The symbolic policy has a succinct form and low complexity which is considered to be more interpretable and easily deployable in real-world settings. [3] and [4] approximate a symbolic policy with genetic programming but are limited to simple tasks with provided or learned world model and suffered from performance decrease compared with NN policies. DSP [5], the state-of-the-art symbolic policy learning method, removes some limitations by employing a recurrent neural network(RNN) as an agent to generate the symbolic policy and trains the RNN with reinforcement learning. However, this method has low data efficiency and requires hundreds of times of environment interactions compared with traditional reinforcement learning algorithms. For environments with multidimensional action spaces, they need a pre-trained NN policy as the anchor model, which brings additional complexity and may limit the final performance. In addition, the complexity of these algorithms makes it difficult to apply them to complex reinforcement learning tasks, e.g. meta-reinforcement learning.

In this paper, we propose an efficient gradient-based learning method called ESPL (Efficient Symbolic Policy Learning) for learning the symbolic policy from scratch in an end-to-end differentiable way. To express the policy in a symbolic form, the proposed ESPL consists of a symbolic network and a path selector. The symbolic network can be considered as a full set of candidate symbolic policies. In the symbolic network, the activation functions are composed of various symbolic operators and the parameters can be regarded as the constants in the symbolic expression. The path selector chooses the proper compact symbolic form from the symbolic network by adaptively masking out irrelevant connections. We design all these modules to be differentiable and represent the policy with a differentiable symbolic expression. Then we can efficiently train the symbolic policy in an off-policy manner and the symbolic policy is directly updated via gradient descent without an additional agent. Experimentally, on several benchmark control tasks, our algorithm is able to produce well-performing symbolic policy while requiring thousands of times fewer environmental interactions than DSP.

Meta-reinforcement learning (meta-RL) is one of the most important techniques for RL applications, which improves the generalization ability on unseen tasks by learning the shared internal structure across several tasks. We raise the question: is it possible to exploit the benefit of the symbolic policy and meta-RL to generate symbolic policies for unseen tasks? While previous symbolic policy methods are too complex to be combined with meta-RL, we combine the proposed ESPL with context-based meta-RL and develop the contextual symbolic policy (CSP). Context-based meta-RL [6, 7, 8, 9] is the most promising meta-RL method which forces the policy to be conditional on context variables that are formed by aggregating experiences. In the CSP framework, the path selector decides the symbolic form based on the context variables. We also involve a parameter generator to generate the constants of the symbolic policy based on the context variables. We build the training process on top of the context-based meta-RL method PEARL [7]. The proposed CSP can generate symbolic policies for unseen tasks given a few trajectories. We find that compared with neural network policies, contextual policies produced by CSP achieve higher generalization performance, efficiency, and show the potential to be interpretable.

The contributions of this paper are three-fold. First, we introduce a novel gradient-based symbolic policy learning algorithm named ESPL that learns the symbolic policy efficiently from scratch. Next, with ESPL we develop the contextual symbolic policy for meta-RL, which can produce symbolic policies for unseen tasks. Finally, we summarize our empirical results which demonstrate the gain of ESPL both in single-task RL and meta-RL. Importantly, we find empirically that contextual symbolic policy improves the generalization performance in PEARL.

## 2 Related Works

### 2.1 Symbolic Policy

The emergence of symbolic policy is partly credited to the development of symbolic regression which is applicable in wide fields, e.g. discovering physics lows [10] and automated CPU design [11]. Symbolic regression aims to find symbolic expressions to best fit the dataset from an unknown fixed function. A series of methods [12, 13, 14] employ genetic programming (GP) to evolve the symbolic expressions. With the development of neural network and gradient descent, some methods [15, 16, 17] involve deep learning for symbolic regression. Some works employ symbolic regression

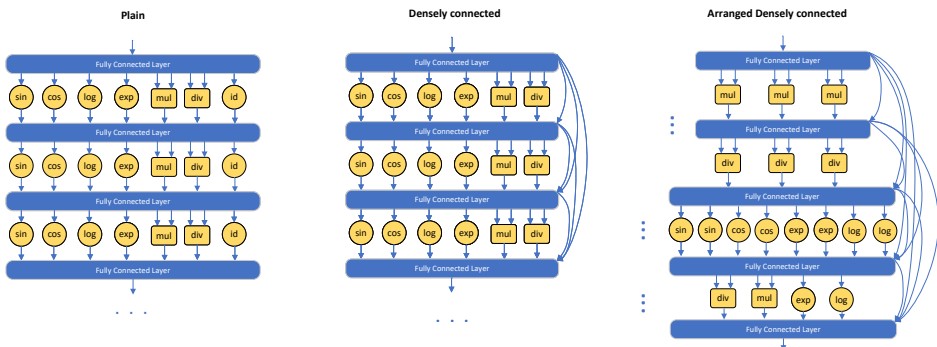

Figure 1: Example network structures for the symbolic network. **Left**: the plain structure. **Middle**: a symbolic work with dense connections. **Right**: a symbolic network with dense connections and arranged operators.

methods to obtain symbolic policies for efficiency and interpretability. [3] and [4] aim to approximate a symbolic policy with genetic programming but require a given dynamics equations or a learned world model. DSP [5], following the symbolic regression method DSR [16], employs a recurrent neural network to generate the symbolic policy. They use the average returns of the symbolic policies as the reward signal and train the neural network with risk-seeking policy gradients. However, for environments with multidimensional action spaces, they need a pre-trained neural network policy as the anchor model. Besides, in this framework, a single reward for reinforcement learning involves many environmental interactions, which is inefficient and makes it hard to combine the symbolic policy with meta-RL. Recently, some works [18, 19] attempt to distill an interpretable policy from a pre-trained neural network policy but have a problem of objective mismatch [5]. Different from the above-mentioned methods, we propose an efficient gradient-based framework to obtain the symbolic policy without any pre-trained model.

## 2.2 Meta-Reinforcement Learning

Meta-RL extends the notion of meta-learning [20, 21, 22] to the context of reinforcement learning. Some works [23, 24, 25] aim to meta-learn the update rule for reinforcement learning. We here consider another research line of works that meta-train a policy that can be adapted efficiently to a new task. Several works [26, 27, 28] learn an initialization and adapt the parameters with policy gradient methods. However, these methods are inefficient because of the on-policy learning process and the gradient-based updating during adaptation. Recently, context-based meta-RL [6, 7, 29] achieve higher efficiency and performance. For example, PEARL [7] proposes an off-policy meta-RL method that infers probabilistic context variables with experiences from new environments. Hyper [29] proposes a hypernetwork where the primary network determines the weights of a conditional network and achieves higher performance. Most of the subsequent context-based meta-RL methods [30, 31, 32] attempt to achieve higher performance by improving the context encoder or the exploration strategy. In this paper, we combine the symbolic policy with meta-RL to form the CSP and consequently improve the efficiency, interpretability and performance of meta-RL. As far as we know, we are the first to learn the symbolic policy for meta-RL.

Our method is also related to some neural architecture search methods and programmatic RL methods. We provide an extended literature review in Appendix E.

## 3 Gradient-based Symbolic Policy Learning

This section introduces the structure of the proposed ESPL, an end-to-end differentiable system. The proposed ESPL consists of two main components: 1) the Symbolic Network, which expresses the policy in a symbolic form, and 2) the Path Selector, which selects paths from the symbolic network to form compact symbolic expressions.

### 3.1 Densely Connected Symbolic Network

To construct a symbolic policy in an end-to-end differentiable form, we propose the densely connected symbolic network $\mathcal{SN}$ as the search space for symbolic policies. Inspired by previous differentiable symbolic regression methods [17, 33], we employ a neural network with specifically designed units, which is named symbolic network. We now introduce the basic symbolic network named plain structure which is illustrated in Figure 1. The symbolic network is a feed-forward network with $L$ layers. Different from traditional neural networks, the activation functions of the symbolic network are replaced by symbolic operators, e.g. trigonometric and exponential functions. For the $l_{th}$ layer of the symbolic network, we denote the input as $x_{l-1}$ and the parameters as weights $w_l$ and biases $b_l$. These parameters serve as the constants in a symbolic expression. We assume that the $l_{th}$ layer contains $m$ unary functions $\{g_1^1, \cdots, g_m^1\}$ and $n$ binary functions $\{g_1^2, \cdots, g_n^2\}$. Firstly, the input of the $l_{th}$ layer will be linearly transformed by a fully-connected layer:

$$y_l = F_l(x_{l-1}) = w_l x_{l-1} + b_l. \tag{1}$$

The fully-connected layer realizes the addition and subtraction in symbolic expressions and produces $m + 2n$ outputs. Then the outputs will go through the symbolic operators and be concatenated to form the layer output:

$$G_l(y_l) = [g_1^1(y_l^1), \cdots, g_m^1(y_l^m), g_1^2(y_l^{m+1}, y_l^{m+2}), \cdots, g_n^2(y_l^{m+2n-1}, y_l^{m+2n})] \tag{2}$$

Then the $l_{th}$ layer of the symbolic network can be formulated as $\mathcal{SN}_l : x_l = G_l(F_l(x_{l-1}))$. Following the last layer, a fully-connected layer will produce a single output. For multiple action dimensions, we construct a symbolic network for each dimension of action.

**Symbolic operator.** The symbolic operators are selected from a library, e.g. $\{sin, cos, exp, log, \times, \div\}$ for continuous control tasks. For the plain structure, we include an identical operator which retains the output of the previous layer to the next layer in the library. To find the symbolic policy via gradient descent, it is critical to ensure the numerical stability of the system. However, this is not natural in a symbolic network. For example, the division operator and the logarithmic operator will create a pole when the input goes to zero and the exponential function may produce a large output. Thus, we regularize the operators and employ a penalty term to keep the input from the "forbidden" area. For example, the logarithmic operator $y = log(x)$ returns $log(x)$ for $x > bound_{log}$ and $log(bound_{log})$ otherwise and the penalty term is defined as $\mathcal{L}_{log} = max(bound_{log} - x, 0)$. The division operator $c = a/b$ returns $a/b$ for $b > bound_{div}$ and 0 otherwise. The penalty term is defined as $\mathcal{L}_{div} = max(bound_{div} - b, 0)$. The details of all regularized operators can be found in the Appendix. To ensure the numerical stability, we involve a penalty loss function $\mathcal{L}_{penalty}$ which is the sum of the penalty terms of all $N$ regularized operators in symbolic networks:

$$\mathcal{L}_{penalty} = \sum_{i=1}^{i=N} \mathcal{L}_{g_i}(x_i). \tag{3}$$

**Dense connectivity.** We introduce dense connections [34] in the symbolic network, where inputs of each layer are connected to all subsequent layers. Consequently, the $l_{th}$ layer of the symbolic network will receive the environment state $s$ and the output of all preceding layers $x_1, \cdots, x_{l-1}$: $x_l = G_l(F_l([s, x_1, \cdots, x_{l-1}]))$. The dense connections improve the information flow between layers and benefit the training procedure. Besides, with these dense skip connections across layers, the combination of symbolic operators becomes more flexible, making the symbolic network more likely to contain good symbolic policies. In addition, we can flexibly arrange the position of operators. For example, if we only arrange the $sin$ operator in the last layer but the oracle expression contains terms like $sin(s_0)$, the input of the $sin$ operator can still be from the original state because of the dense connections. We give an example of arranged operators in Figure 1 which we use for all tasks in the experiments. In this symbolic network, we heuristically involve more multiplication and division operators at shallow layers to provide more choice of input processed by simple operators for complex operations such as sines and cosines.

### 3.2 The Path Selector

The symbolic network serves as a full set of the search space of symbolic expressions. To select the proper paths from the symbolic network to produce a compact symbolic policy, we reduce the

number of paths involved in the final symbolic policy then proper paths remain and redundant paths are removed. This can be naturally realized by minimizing the $L_0$ norm of the symbolic network weights. As the $L_0$ norm is not differentiable, some methods [17, 33] employ $L_1$ norm instead of $L_0$ norm. However, $L_1$ will penalize the magnitude of the weights and result in performance degradation. Inspired by the probability-based sparsification method [35, 36, 37], we propose a probabilistic path selector which selects paths from the network by multiplying a binary mask on the weights of the symbolic network $\boldsymbol{w}$. The binary mask $m_i$ is sampled from the Bernoulli distribution: $m_i \sim Bern(p_i)$, where $p_i \in [0, 1]$ serves as the probability. Then the final weights of the symbolic network are $\boldsymbol{w_m} = \boldsymbol{w} \bigotimes \boldsymbol{m}$, where $\bigotimes$ is the element-wise multiply operation. Consequently, to get a compact symbolic expression, we only need to minimize the expectation of the $L_0$ norm of the binary mask $\mathbb{E}_{\boldsymbol{m} \sim Bern(\boldsymbol{m}|\boldsymbol{p})} \|\boldsymbol{m}\|_0 = \sum p_i$, without penalizing the magnitude of the weights. During the process of collecting data or testing, we can directly sample the binary mask from the Bernoulli distribution. Then we can obtain the symbolic policy $\pi_{sym}$ by removing paths with zero weight and simplifying the symbolic expression.

However, the sampling process does not have a well-defined gradient. Thus, for the training process we build up our sampling function with the gumbel-softmax trick [38]. As the mask $\boldsymbol{m}$ is binary categorical variables, we replace the *softmax* with *sigmoid* and named the sampling function as *gumbel sigmoid*. The *gumbel sigmoid* function can be formulated as:

$$\boldsymbol{m_{gs}} = sigmoid(\frac{log(\frac{\boldsymbol{p}}{1-\boldsymbol{p}}) + \boldsymbol{g_1} - \boldsymbol{g_0}}{\tau}), \tag{4}$$

where $\boldsymbol{g_1}$ and $\boldsymbol{g_0}$ are i.i.d samples drawn from $Gumbel(0, 1)$. $\tau$ is the temperature annealing parameter. Note that $\boldsymbol{m_{gs}}$ is still not a binary mask. To obtain a binary mask but maintain the gradient, we employ the Straight-Through (ST) trick: $\boldsymbol{m} = \mathbb{1}_{\geq 0.5}(\boldsymbol{m_{gs}}) + \boldsymbol{m_{gs}} - \overline{\boldsymbol{m_{gs}}}$, where $\mathbb{1}_{\geq 0.5}(x) \in \{0, 1\}^n$ is the indicator function and the overline means stopping the gradient. During training, we do not remove paths with zero weight and directly use symbolic network $\mathcal{SN}(\boldsymbol{w_m})$ as the policy.

We also involve a loss function $\mathcal{L}_{select}$ to regularize the sum of probabilities $\boldsymbol{p}$ which is the expectation of the $L_0$ norm of the binary mask $\boldsymbol{m}$. To limit the minimum complexity of symbolic policies, we involve the minimum $L_0$ norm defined as $l_{min}$. Then the loss function can be defined as:

$$\mathcal{L}_{select} = max(\sum p_i - l_{min}, 0). \tag{5}$$

### 3.3 Implementation

In practice, we build our off-policy learning framework on top of the soft actor-critic algorithm (SAC) [39]. We employ the neural network $Q(s, a)$ parameterized by $\theta_Q$ as the critic (state-action-value function). To construct a stochastic policy, we also employ a small neural network $F(s)$ parameterized by $\theta_F$ to output the standard deviation. Note that $Q(s, a)$ and $F(s)$ are only used during training. We optimize the weights $\boldsymbol{w}$, the biases $\boldsymbol{b}$ of the symbolic network, the probabilities $\boldsymbol{p}$ in the path selector, and $\theta_F$ with the combination of actor loss from SAC, $\mathcal{L}_{penalty}$ and $\mathcal{L}_{select}$. We update $\theta_Q$ with the critic loss from SAC. During training, we decrease the temperature parameter $\tau$ of *gumbel sigmoid* linearly and decrease the $l_{min}$ from the count of the original parameters $\boldsymbol{w}$ to a target value with a parabolic function. We summarize the training procedure and give the pseudo-code in Appendix C.

## 4 Contextual Symbolic Policy for Meta-RL

### 4.1 Background

In the field of meta-reinforcement learning (meta-RL), we consider a distribution of tasks $p(\kappa)$ with each task $\kappa \sim p(\kappa)$ modeled as a Markov Decision Process(MDP). In common meta-RL settings, tasks share similar structures but differ in the transition and/or reward function. Thus, we can describe a task $\kappa$ with the 6-tuple $(\mathcal{S}, \mathcal{A}, \mathcal{P}_\kappa, \rho_0, r_\kappa, \gamma)$. In this setting, $\mathcal{S} \subseteq \mathbb{R}^n$ is a set of n-dimensional states, $\mathcal{A} \subseteq \mathbb{R}^m$ is a set of m-dimensional actions, $\mathcal{P}_\kappa : \mathcal{S} \times \mathcal{A} \times \mathcal{S} \to [0, 1]$ is the state transition probability distribution, $\rho_0 : \mathcal{S} \to [0, 1]$ is the distribution over initial states, $r_\kappa : \mathcal{S} \times \mathcal{A} \to \mathbb{R}$ is the reward function, and $\gamma \in (0, 1)$ is the per timestep discount factor. Following the setting of prior works [7, 8], we assume there are $M$ meta-training tasks $\{\kappa_m\}_{m=1,\cdots,M}$ sampled from the training

Table 1: Symbolic policies produced by ESPL.

| Environment | ESPL |
|---|---|
| CartPole | $a_1 = 17.17s_3 + 1.2s_4$ |
| MountainCar | $a_1 = 8.06sin(9.73s_2 - 0.18) + 1.26$ |
| Pendulum | $a_1 = -(4.27s_1 + 0.62)(1.9s_2 + 0.42s_3)$ |
| InvDoublePend | $a_1 = 12.39s_5 - 4.48sin(0.35s_2 + 4.51s_5 + 1.23s_6 + 7.97s_8 + 1.23s_9 + 0.08) + 0.34$ |
| InvPendSwingup | $a_1 = 4.33sin(0.17 * s_1 + 0.14s_2 + 0.49s_3 + 1.76s_4 + 0.33s_4 - 0.29) - 0.65$ |
| LunarLander | $a_1 = (0.14 - 2.57s_4)(0.48 - 0.68log(0.5s_2)) - 1.44$ 
 $a_2 = -5.72s_3 + 4.42sin(2.54s_5 + 0.03) - 0.4 - \frac{-6.5s_6 - 2.13cos(0.78sin(4.15s_1 - 0.05) + 1.98) - 0.98}{4.71*s7 + 0.77}$ |
| Hopper | $a_1 = -0.32s_{12} - 1.46s_8 - 0.83s_{10} - 0.11sin(0.26s_{11} - 5s_{13} - 2.57s_6 + 0.38) - 0.92$ 
 $a_2 = -0.52s_{12} - 3.63s_4 - 4.58s_8 + 0.68exp(-7.31s_{11} - 2.5s_{13}) + 0.58 + \frac{-1.62s_6 + 3.89s_9 - 4.7}{1.33 - 0.44s_{13}}$ 
 $a_3 = 0.83 + \frac{1.12s_1 - 0.47 - 0.1exp((10.05s_1 - 1.76s_6 + 1.65)(0.22s_{13} - 1.88s_{14} + 1.32))(5.59*s_1 - 0.08)}{0.23 + 0.21exp((10.05s_1 - 1.76s_6 + 1.48)(0.22s_{13} - 1.88s_{14} + 1.32))}$ |
| BipedalWalker | $a_1 = 1.45 - 2.94cos(-0.73s_5 + (0.06 - 1.06s_3)(-1.33s_{12} - 0.28s_6 + 0.41) + 1.32)$ 
 $a_2 = 7.53exp(0.4s_1 - 0.13s_6 - 0.52sin(1.5s_7 - 0.24)) - 11.1$ 
 $a_3 = -\frac{1.07s_6 + 0.41}{4.56s_9 + (0.2 - 3.01s_{21})(-1.8s_1 - 0.03s_7 - 0.96) - 0.54} + 0.55$ 
 $a_4 = -0.28 + \frac{-3.32s_{12} + 5.64s_3 + 0.29s_{22} - 2.46}{3.26s_{23} - 1.45}$ |

Table 2: Performance comparison of symbolic policies and neural policies for seven different DRL algorithms.

| Environment | DDPG | TRPO | A2C | PPO | ACKTR | SAC | TD3 | Regression | DSP | ESPL |
|---|---|---|---|---|---|---|---|---|---|---|
| Cartpole | 1000 | 1000 | 1000 | 1000 | 1000 | 1000 | 1000 | 211.82 | 1000 | 1000 |
| Mountaincat | 95.36 | 93.6 | 93.97 | 93.76 | 93.79 | 94.68 | 93.87 | 95.16 | 99.11 | 94.02 |
| Pendulum | -155.6 | -145.49 | -157.59 | -160.14 | -201.57 | -154.82 | -155.06 | -1206.9 | -155.4 | -151.72 |
| InvDoublePend | 9347.1 | 9188.43 | 9359.81 | 9356.59 | 9359.06 | 9359.92 | 9359.25 | 637.2 | 9149.9 | 9359.9 |
| InvPendSwingup | 891.48 | 892.9 | 254.71 | 890.1 | 890.11 | 891.32 | 892.25 | -19.21 | 891.9 | 890.36 |
| LunarLander | 266.05 | 265.26 | 238.51 | 269.65 | 271.53 | 276.92 | 272.13 | 56.08 | 261.36 | 283.56 |
| Hopper | 1678.84 | 2593.56 | 2104.98 | 2586.56 | 2583.88 | 2613.16 | 2743.9 | 47.35 | 2122.4 | 2442.48 |
| BipedalWalker | 209.42 | 312.14 | 291.79 | 287.43 | 309.57 | 308.31 | 314.24 | -110.77 | 311.78 | 309.43 |
| Worst Rank | 9 | 10 | 9 | 9 | 9 | **6** | 7 | 10 | 9 | **6** |
| Average Rank | 5.5 | 4.125 | 6.25 | 6.125 | 5.375 | 3 | **2.875** | 9.125 | 4.625 | 3.5 |

tasks distribution $p_{train}(\kappa)$. For meta-testing, the tasks are sampled from the test tasks distribution $p_{test}(\kappa)$. The two distributions are usually the same in most settings but can be different in out-of-distribution(OOD) settings. We denote context $c_T = \{(s_1, a_1, s_1', r_1), \cdots, (s_T, a_T, s_T', r_T)\}$ as the collected experiences. For context-based meta-RL, the agent encodes the context into a latent context variable $z$ with a context encoder $q(z|c)$ and the policy $\pi$ is conditioned on the current state and the context variable $z$. During adaptation, the agent first collects experiences for a few episodes and then updates the context variables. Then the contextual policy is able to adapt to new tasks according to the context variables. The meta-RL objective can be formulated as $\max_{\pi} \mathbb{E}_{\kappa \sim p(\kappa)}[\mathbb{E}_{c_T \sim \pi}[R(\kappa, \pi, q(z|c_T))]]$, where $R(\kappa, \pi, q(z|c_T))$ denotes the expected episode return.

## 4.2 Incorporating the Context Variables

To quickly adapt to new tasks, we need to incorporate the context variables $z \sim q(z|c_\kappa)$ to the symbolic network and produce different symbolic policies for different tasks $\kappa$ sampled from the task distribution $p(\kappa)$. To condition the parameters of the symbolic expression on the context variable, we propose a parameter generator: $\boldsymbol{w}, \boldsymbol{b} = \Phi(z)$ which is a neural network to produce the parameters of symbolic networks for all action dimensions based on the context variables. We also involve a neural network to generate the probabilities of the path selector: $\boldsymbol{p} = \Psi(z)$. Then the contextual symbolic network can generate different symbolic expression forms according to the context variables.

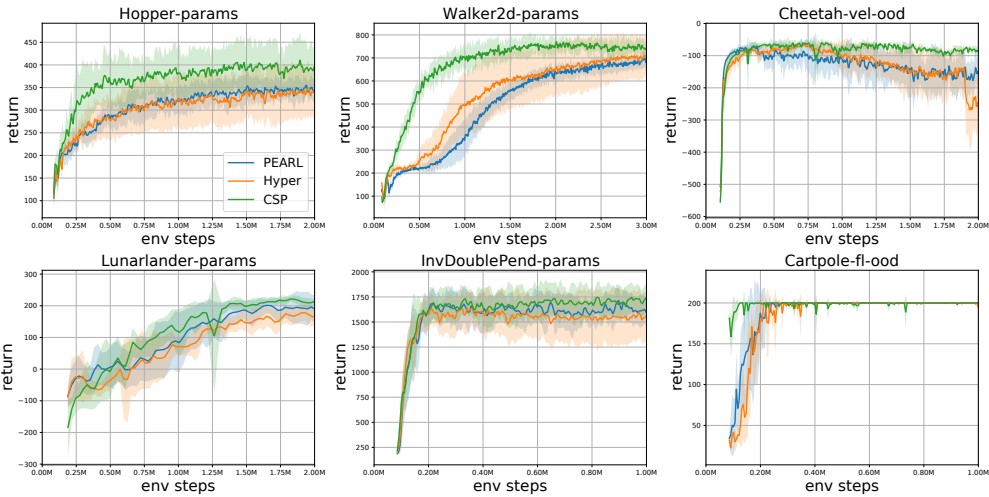

Figure 2: Comparison for different kinds of contextual policies on meta-RL tasks. We show the mean and standard deviation of returns on test tasks averaged over five runs.

## 4.3 Training Schedule

We train the CSP in an off-policy manner. For meta-training epoch $t$, the agent first collects experiences of different training tasks into the corresponding buffer $\mathcal{B}_{\kappa_i}$ for several iterations. At the beginning of each data collection iteration, we sample context $c_T$ from buffer $\mathcal{B}_{\kappa_i}$ and sample context variables $z \sim q(z|c_T)$ as PEARL [7] does. The difference is that we also sample the symbolic policy with $\Phi(z)$ and $\Psi(z)$ and use the sampled policy for the following steps of the iteration. For training, we sample RL batch and context from the buffer and optimize the context encoder $q(z|c_T)$ to recover the state-action value function. For each training step, we sample a new symbolic policy. We employ the soft actor-critic to optimize the state-action value function. For the parameter generator and the path selector, we employ $\mathcal{L}_{select}$ and $\mathcal{L}_{penalty}$ in addition to the SAC loss. During training, we decrease the temperature parameter $\tau$ and $l_{min}$ just like single-task RL. More details and the pseudo-code can be found in Appendix C.

# 5 Experiment

## 5.1 Experimental Settings

**Environment.** For single-task RL, we evaluated our method on benchmark control tasks which are presented in DSP: (1) CartPole; (2) MountainCar; (3) Pendulum; (4) InvertedDoublePendulum; (5) InvertedPendulumSwingup; (6) LunarLander; (7) Hopper; (8) BipedalWalker. For meta-RL, we evaluate the CSP on several continuous control environments which are modified from the environments of OpenAI Gym [40] to be meta-RL tasks similar to [7, 9, 30]. These environments require the agent to adapt across dynamics (random system parameters for Hopper-params, Walker2d-params, Lunarlander-params, InvDoublePend-params, different force magnitude and pole length for Cartpole-fl-ood) or reward functions (target velocity for Cheetah-vel-ood).

**Methods.** In the single-task RL experiments, for the neural network policies, we compare our method with seven state-of-the-art DRL algorithms: DDPG, TRPO, A2C, PPO, ACKTR, SAC, and TD3 [41, 42, 43, 44, 45, 39, 46]. The results are obtained with the tuned pretrained policies from an open-source repository Zoo [47]. For symbolic policies, we include the Regression method and DSP. The Regression policies are produced with two steps: 1) generate a dataset of observation action trajectories from the best pre-trained policy from Zoo; 2) perform deep symbolic regression [16] on the dataset and select the expression with the lowest error for each action. DSP first trains a recurrent neural network with reinforcement learning to produce symbolic policies and then optimizes the constant with several methods such as Bayesian Optimization [48].

Table 3: The number of environment episodes used for learning symbolic policies.

| Environment | Regression | DSP | ESPL |
|---|---|---|---|
| CartPole | 1000 | 2M | 500 |
| MountainCar | 1000 | 2M | 500 |
| Pendulum | 1000 | 2M | 500 |
| InvDoublePend | 1000 | 2M | 500 |
| InvPendSwingup | 1000 | 2M | 500 |
| LunarLander | 1000 | 0.4M | 1000 |
| Hopper | 1000 | 0.4M | 3000 |
| BipedalWalker | 1000 | 0.4M | 2000 |

**Evaluation.** Following DSP, we evaluate the proposed ESPL by averaging the episodic rewards across 1,000 episodes with different environment seeds. The evaluation for all the baselines is also in accordance with this protocol. For the DSP and regression method, we use the results from the DSP paper. DSP performs 3n independent training runs for environments with n-dimension action and select the best symbolic policy. For a fair comparison, we perform three independent runs and select the best policy for ESPL and DRL methods. For meta-RL, we run all environments based on the off-policy meta-learning framework proposed by PEARL [7] and use the same evaluation settings. We compare CSP with PEARL which concatenates the observation and context variables as the input of policy and Hyper [29] which generate the parameters of policy with a ResNet model based on the context variables. Note that the original Hyper also modifies the critic, but we build all the critics with the same network structure for consistency. More details of the experimental settings can be found in Appendix D.

## 5.2 Comparisons for Single-task RL

**Performance.** In Table 2, we report the average episode rewards of different algorithms. Among symbolic policies, the proposed ESPL achieves higher or comparable performance compared with DSP while the regression method performs poorly in most environments. To directly compare ESPL with other algorithms across different environments, we rank the algorithms and calculate the average rank and worst-case rank. We find that TD3 and SAC outperform other algorithms. For symbolic policy methods, the proposed ESPL achieves superior performance compared to DSP. ESPL is also comparable with the better-performing algorithms of DRL.

**Data efficiency.** In Table 3, we report the number of environment episodes required for learning the symbolic policy in different algorithms. The proposed ESPL requires fewer environmental interactions to learn the symbolic policies in most environments. Compared with DSP, the proposed ESPL uses thousands of times fewer environment episodes.[2] Although the regression method requires a similar number of environment episodes as ESPL, it fails to find meaningful policies in most environments according to Table 2. Besides, the proposed ESPL is trained from scratch while the regression method and DSP need pretrained neural network policies. In conclusion, the proposed ESPL greatly improves the data efficiency of symbolic policy learning.

## 5.3 Comparisons for Meta-RL

**Performance.** For meta-RL tasks, we report the learning curves of undiscounted returns on the test tasks in Figure 2. We find that CSP achieves better or comparable performance in all the environments compared with previous methods. In Hopper-params, Walker2d-params and InvDoublePend-params, CSP outperforms PEARL and Hyper during the whole training process. In Lunarlander-params, CSP achieves better final results. In Cartpole-fl-ood, CSP adapts to the optimal more quickly. In the out-of-distribution task Cheetah-vel-ood, we find the performance of PEARL and Hyper decrease during training because of over-fitting. But our CSP is less affected. Thus, expressing the policy in the symbolic form helps improve the generalization performance.

**Deploying efficiency.** We also evaluate the deploying efficiency of contextual policies. We first calculate the flops of each kind of policy per inference step. Then we consider an application scenario

---

[2]For DSP, we only record the environment episodes required for reinforcement learning as the exact number of iterations for constant optimization is not provided in DSP.

Table 4: FLOPs (k)/Inference time (ms) of different contextual policies.

| Environment | CSP | PEARL | Hyper |
|---|---|---|---|
| Walker2d-params | **3.11/20.9** | 189.3/27.0 | 5.64/22.6 |
| Hopper-params | **0.51/4.13** | 186.9/26.6 | 4.1/17.2 |
| InvDoublePend-params | **0.039/0.37** | 186.0/25.1 | 3.59/12.3 |
| Cartpole-fl-ood | **0.004/0.042** | 183.9/23.9 | 1.79/9.08 |
| Lunarlander-g | **0.015/0.l4** | 185.4/23.4 | 3.08/12.3 |
| Cheetah-vel-ood | **0.53/4.9** | 190.2/28.4 | 7.18/24.2 |

that the algorithm control five thousand simulated robots with the Intel(R) Xeon(R) Gold 5218R @ 2.10GHz CPU and record the elapsed time per inference step. We report the results in Table 4. Compared to PEARL, CSP reduces the FLOPs by 60-45000x and reduces the inference time by up to 600x. Compared to Hyper, CSP reduces the flops by 2-450x and reduces the inference time by up to 200x. Thus, compared with pure NN policies, the CSP has a significant advantage in computational efficiency.

## 5.4 Analysis of symbolic policies

**Interpretability.** For single-RL tasks, we report the symbolic policies found by ESPL for each environment in Table 1. The policies in the symbolic form are simple and we can directly glean insight from the policies by inspection, or in other words, interpretability. For example, the goal of LunarLander is to land a spacecraft on a landing pad without crashing. The action $a_1$ controls the main engine and $a_2$ controls the orientation engines. In our symbolic policy, $a_1$ is a function of $s_1$ (the height) and $s_4$ (the vertical velocity). The main engine turns on to counteract downward motion when the height is low. Action $a_2$ is a combination of a term about $s_3$(the horizontal velocity), a term about $s_5$(the angle), and a highly nonlinear term about $s_6$ (the angular velocity), $s_1$ (the horizontal position) and $s_7$ (whether the leg has landed). Thus, the policy adjusts the orientation engines based on the incline and horizontal motion to move the spacecraft to the center of the landing pad. In meta-RL, the CSP also shows the potential to be interpretable. We take the Cartpole-fl-ood environment as an example and illustrate the Cartpole system in Figure 3. The form of the symbolic policies produced by CSP is $action = c1 * \theta + c_2 * \dot{\theta} + b$, where $c1$ and $c2$ are the positive coefficients and $b$ is a small constant which can be ignored. Then the policy can be interpreted as pushing the cart in the direction that the pole is deflected or will be deflected. To analyze the difference between policies for different tasks, we uniformly set the force magnitude and the length of the pole. Then we generate the symbolic policy with CSP and record the coefficients. As Figure 4 shows, $c1$ and $c2$ tend to increase when the force magnitude decrease and the length increase, which is in accord with our intuition. We also provide the human study results of the interpretability in Appendix F.3.

**Complexity.** we compare the length of the symbolic policies and define $length = \frac{\sum_{i=1}^{i=n} N_o^i + N_c^i + N_v^i}{n}$, where $n$ is the dimension of the action, $i$ is the index of the action dimension, $N_o^i$ is the number of operators, $N_c^i$ is the number of constant terms, $N_v^i$ is the number of variable terms. We give a comparison of the length of the symbolic policies in Table 5.

Table 5: Comparison of the length of the symbolic policies.

| | Average | CartPole | MountainCar | Pendulum | InvDoublePend | InvPendSwingup | LunarLander | Hopper | BipedalWalker |
|---|---|---|---|---|---|---|---|---|---|
| **ESPL** | 12.91 | 3 | 6 | 7 | 15 | 13 | 16.5 | 24.6 | 17 |
| **DSP** | 8.25 | 3 | 4 | 8 | 1 | 19 | 6.5 | 12 | 12.5 |

In the benchmark environments used in the literature, in some environments ESPL produces longer symbolic policies than DSP, in others ESPL produces similar or shorter symbolic policies than DSP. In general, symbolic policies produced by ESPL are only slightly longer than the symbolic policies produced by DSP.

## 5.5 Ablation

Finally, we carry out experiments by ablating the features of the proposed ESPL. We change the structure of the symbolic network and replaced the path selector with the $L_1$ norm minimization. We

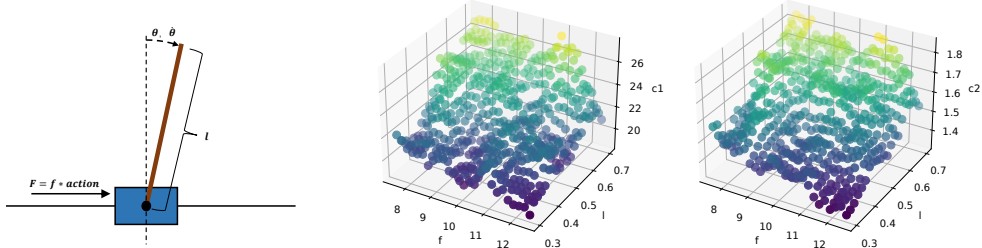

Figure 3: The Cartpole system to be controlled.

Figure 4: The coefficients of symbolic policies for Cartpole environments with different force magnitude and pole length.

Table 6: Ablation results for single-task RL. $ESPL_p$ and $ESPL_d$ means replacing the symbolic network with a plain structure and a densely connected structure respectively. $ESPL_{l_1}$ means replacing the path selector with the $L_1$ norm minimization.

| Environment | ESPL | $ESPL_p$ | $ESPL_d$ | $ESPL_{l_1}$ |
|---|---|---|---|---|
| CartPole | **1000** | **1000** | **1000** | **1000** |
| MountainCar | 94.02 | 93.69 | 93.83 | **94.17** |
| Pendulum | -151.72 | -183.16 | **-144.08** | -163.54 |
| InvDoublePend | **9359.9** | 9357.6 | 9197.44 | 8771.18 |
| InvPendSwingup | **890.36** | 844.84 | 890.01 | 865.38 |
| LunarLander | **283.56** | 263.95 | 277.36 | 271.69 |
| Hopper | **2442.48** | 2003.24 | 2316.54 | 1546.35 |
| BipedalWalker | **309.43** | -11.63 | 298.50 | 6.81 |

report the average episode rewards for single-task RL in Table 6. As the experiment results show, without the path selector or the dense connections, the performance degrades, especially in Hopper and BipedalWalker. With the path selector or the dense connections, $ESPL_d$ is able to perform well for all the environments while we observe that the arranged symbolic operators can further improve the performance. We also provide the ablation study results for meta-RL in Appendix F.2.

## 6 Conclusions

In this paper, we introduce ESPL, an efficient gradient-based symbolic policy learning method. The proposed ESPL is able to learn the symbolic policy from scratch in an end-to-end way. The experiment results on eight continuous control tasks demonstrate that the approach achieves comparable or higher performance than both NN-based policies and previous symbolic policies while greatly improving the data efficiency compared with the previous symbolic policy method. We also combine our method with meta-RL to generate symbolic policies for unseen tasks. Empirically, compared with neural network policies, the proposed symbolic policy achieves higher performance and efficiency and shows the potential to be interpretable. We hope ESPL can inspire future works of symbolic policy for reinforcement learning or meta-reinforcement learning. Besides, as the symbolic policy is a white box and more dependable, the proposed ESPL may promote applications of reinforcement learning in industrial control and automatic chip design.

**Limitations and future work.** In this paper, we focus on continuous control tasks with low-dimensional state space. The proposed ESPL and CSP can not directly generate a symbolic policy for tasks with high-dimensional observation like images. A possible method is to employ a neural network to extract the environmental variables and generate symbolic policy based on these environmental variables. We leave this in the future work. Symbolic policies generally have good interpretability. However, when the task is too complex, the symbolic policy is also more complex, making the interpretability decrease. Solving this problem is also an important direction for future work. For application, we will further learn a symbolic policy for automated CPU design based on this framework to optimize the performance/power/area (PPA) of the CPU.

# 7 Acknowledgement

This work is partially supported by the National Key R&D Program of China (under Grant 2021ZD0110102), the NSF of China (under Grants 61925208, 62102399, 62222214, 62002338, U22A2028, U19B2019), CAS Project for Young Scientists in Basic Research (YSBR-029), Youth Innovation Promotion Association CAS and Xplore Prize.

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

# A Symbolic Operators

To ensure the numerical stability of the proposed symbolic learning framework, we regularize the operators and employ a penalty term to keep the input from the "forbidden" area. We show the operators and the corresponding penalty terms as follows:

- Multiplying operator:

$$y = min(max(x_1, -100), 100) * min(max(x_2, -100), 100),$$

  the penalty term can be formulated as:

$$L_{mul} = max(x_1 - 100, 0) + max(-100 - x_1, 0) + max(x_2 - 100, 0) + max(-100 - x_2, 0)$$

- Division operator:

$$y = \begin{cases} 0, & x_2 < 0.01, \\ \dfrac{x_1}{x_2}, & x_2 \geq 0.01. \end{cases}$$

  the penalty term can be formulated as:

$$L_{div} = max(0.01 - x_2, 0)$$

- Sine operator: $y = sin(x)$, the penalty term is set as zero: $L_{sin} = 0$.

- Cosine operator: $y = cos(x)$, the penalty term is set as zero: $L_{cos} = 0$.

- Exponential operator:

$$y = exp(min(max(x, -10), 4)),$$

  the penalty term can be formulated as:

$$L_{exp} = max(x - 4, 0) + max(-10 - x, 0)$$

- Log operator:

$$y = log(max(x, 0.001)),$$

  the penalty term can be formulated as:

$$L_{log} = max(0.001 - x, 0)$$

- Identity operators: $y = x$, the penalty term is set as zero: $L_{identity} = 0$.

- Condition operator: $y = sigmoid(x_1) * x_2 + (1 - sigmoid(x_1)) * x_3$, the penalty ter is set as zero: $L_{condition} = 0$.

In practice, the identity operator is only used in the plain structure. For all the environments in single-task RL and meta-RL, we use the symbolic network structure described in Section 3.1. Especially, for the meta-RL environment Cheetah-vel-ood, we add one Condition operator in each layer. We think the Condition operator will be useful for environments where the reward function changes. During training, we involve a penalty loss function $\mathcal{L}_{penalty}$ which is the sum of the penalty terms of regularized operators in symbolic networks:

$$\mathcal{L}_{penalty} = \sum_{i=1}^{i=N} \mathcal{L}_{g_i}(x_i). \tag{6}$$

We show the learning curves of the penalty loss function for single-task RL in Figure 5 and for meta-RL in Figure 6. During the training process, for all environments in both single-task RL and meta-RL, the penalty loss function remains on a very small order of magnitude, which indicates that most of the operators in the symbolic network work the same as the original unregularized operators.

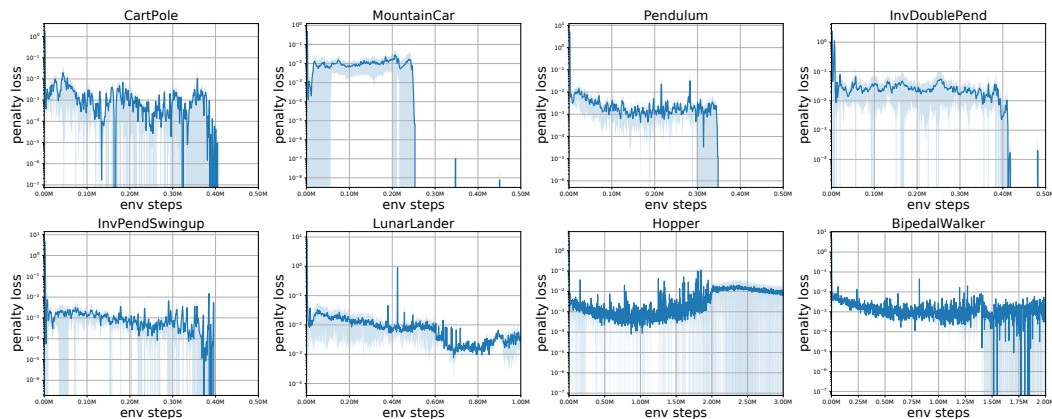

Figure 5: Learning curves of the penalty loss function in single-task RL. The shaded area spans one standard deviation.

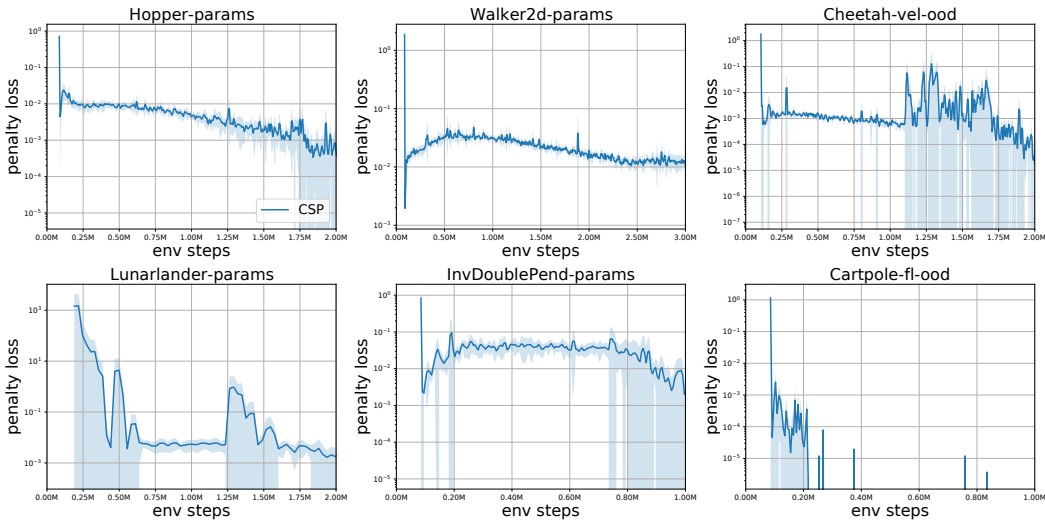

Figure 6: Learning curves of the penalty loss function in meta-RL.

# B  Environment details.

## B.1  Single-task RL

For single-task RL, we use the same benchmark control tasks as DSP [5]. We describe these environments as follows:

- CartPole: In the environment, there is a cart that can move linearly with a pole fixed on it. The goal is to balance the pole by applying forces in the left and right direction on the cart. The max horizon length is set as 1000. We use a continuous version from an open-source implementation `https://gist.github.com/iandanforth`.

- MountainCar: In the environment, there is a car placed stochastically at the bottom of a sinusoidal valley. The action is the accelerating the car in either direction. The goal of the environment is to accelerate the car to reach the goal state on top of the right hill. We use MountainCarContinuous-v0 from OpenAI Gym[40].

- Pendulum: In this environment, there is a pendulum attached to a fixed point at one end. The pendulum starts in a random position and the goal is to apply torque on the free end to swing it into an upright position. We use Pendulum-v0, from OpenAI Gym.

---

**Algorithm 1** The training process of ESPL.

---

**Input**: The number of iterations for temperature and $L_0$ norm schedule $t_s$. The target temperature $\tau_t$ and target minimum $L_0$ norm $l_t$. The symbolic network $\mathcal{SN}$ and the parameters $\boldsymbol{w}, \boldsymbol{b}$. The probabilities $\boldsymbol{p}$ of the path selector.

1: Initialize replay buffers $\mathcal{B}$.
2: **for** training iteration $t = 0$ to $T - 1$ **do**
3:     $\tau = (1 - \tau_t) * (1 - \frac{min(t, t_s)}{t_s}) + \tau_t$
4:     Generate symbolic policy $\pi_{sym}$ with the symbolic network and the path selector.
5:     **for** environment step $k = 0$ to $K - 1$ **do**
6:         Collect data with $a \sim \mathcal{N}(\pi_{sym}(s), F(s))$ and add to buffer $\mathcal{B}$
7:     **end for**
8:     $l_{min} = l_t + (1 - l_t) * \left(1 - \frac{min(t, t_s)}{t_s}\right)^2$
9:     **for** steps in training step **do**
10:        Sample RL Batch from the buffer $\mathcal{B}$
11:        Sample $\boldsymbol{m}$ with *gumbel sigmoid* and ST trick.
12:        Obtain the policy: $\mathcal{SN}(\boldsymbol{w_m})$
13:        Calculate loss for the Critic: $\mathcal{L}_{critic} = \mathcal{L}_{critic}^{sac}$
14:        Calculate loss for the Actor:
        $\mathcal{L}_{actor} = \mathcal{L}_{actor}^{sac} + \alpha_1 \mathcal{L}_{penalty} + \alpha_2 \mathcal{L}_{select}$
15:     **end for**
16:     Update the $\theta_Q$ with $\mathcal{L}_{critic}$
17:     Update $\boldsymbol{w}, \boldsymbol{b}, \boldsymbol{p}, \theta_F$ with $\mathcal{L}_{actor}$
18: **end for**

---

- InvertedDoublePendulum: In this environment, there is a cart that can move linearly, with a pole fixed on it and a second pole fixed on the other end of the first one. The cart can be pushed left or right, and the goal is to balance the second pole on top of the first pole by applying continuous forces on the cart. We use InvertedDoublePendulumBulletEnv-v0 from PyBullet [49].

- InvertedPendulumSwingup: This environment is a combination of CartPole and Pendulum. The goal of the environment is to swing-up the pendulum from its natural pendent position to its inverted position. We use InvertedPendulumSwingupBulletEnv-v0 from PyBullet.

- LunarLander: In this environment, there is a spacecraft with a main engine and two orientation engines. The goal of the environment is to land a spacecraft on a landing pad without crashing by controlling the engines. We use LunarLanderContinuous-v2 from OpenAI Gym.

- Hopper: In this environment, there is a two-dimensional one-legged robot that should move forward. We use HopperBulletEnv-v0 from PyBullet.

- BipedalWalker: This is a simple 4-joint walker robot environment. The goal is to control the walker to move forward. We use BipedalWalker-v2 from OpenAI Gym.

## B.2 Meta-RL

We introduce the details of the environments used in the meta-RL experiments. We build all the meta-RL environments by modifying the origin environments of OpenAI Gym[3]. We describe these environments as follows:

- Walker2d-params: This environment is built by modifying the OpenAI gym Walker2d, in which a two-dimensional two-legged robot should move forward. For each task, the dynamics parameters of the environment are randomly initialized. The horizon length is set as 200. For the experiment, we sample 40 training tasks and 10 test tasks.

- Cheetah-vel-ood: This environment is built by modifying the OpenAI gym Half Cheetah. In the modified environment, a 2-dimensional robot with nine links should achieve a target

---

[3]We build our meta-RL environments based on the random-param-env https://github.com/dennisl88/rand_param_envs.

**Algorithm 2** The training process of CSP.

**Input**: Batch of training tasks $\{\kappa_i\}_{i=1,...m}$. The number of iterations of temperature and target $L_0$ norm schedule $t_s$. Target temperature $\tau_t$. The target $L_0$ norm of the mask at the end of training $l_t$. The scale of the penalty loss $\alpha_1$. The scale of the loss to regularize the sum of probabilities $\alpha_2$. The scale of the loss for the KL divergence $\beta$.

1: Initialize replay buffers $\mathcal{B}_i$ for each training tasks.
2: **for** training iteration $t = 0$ to $T - 1$ **do**
3:     $\tau = (1 - \tau_t) * (1 - \frac{min(t,t_s)}{t_s}) + \tau_t$
4:     **for** each task $\kappa_i$ **do**
5:         Initialize context $c_i = \{\}$
6:         **for** $k = 0$ to $K - 1$ **do**
7:             Sample $z \sim q_{\theta_q}(z|c_i)$
8:             Generate symbolic policy $\pi_{sym}$ with $\Phi(z)$ and $\Psi(z, \tau)$
9:             Collect data with $a \sim \mathcal{N}(\pi_{sym}(s), F(s, z))$ and add to buffer $\mathcal{B}_i$.
10:            Update $c_i = \{(s_j, a_j, s'_j, r_j)\}_{j=1,\cdots,N} \sim \mathcal{B}_i$
11:         **end for**
12:     **end for**
13:     **for** steps in training step **do**
14:         Sample a batch of tasks.
15:         **for** each $\kappa_i$ in the batch **do**
16:             Sample context $c_i$ and RL Batch $b_i$ from the buffer $\mathcal{B}_i$
17:             Sample context variables $z \sim q_{\theta_q}(z|c_i)$
18:             Obtain parameters $\boldsymbol{w}, \boldsymbol{b}$ with $\Phi(z)$
19:             Obtain probabilities $\boldsymbol{p}$ with $\Psi(z)$
20:             Sample $\boldsymbol{m}$ with *gumbel sigmoid* and ST trick.
21:             Obtain the policy: $\mathcal{SN}(\boldsymbol{w_m})$
22:             $l_{target} = l_t + (1 - l_t) * \left(1 - \frac{min(t,t_s)}{t_s}\right)^2$
23:             Calculate loss for the critic: $\mathcal{L}^i_{critic} = \mathcal{L}^{sac}_{critic}$
24:             Calculate loss for the Actor:
                $\mathcal{L}^i_{actor} = \mathcal{L}^{sac}_{actor} + \alpha_1 \mathcal{L}_{penalty} + \alpha_2 \mathcal{L}_{select}$
25:             Calculate the KL divergence $\mathcal{L}^i_{KL} = D_{KL}(q_{\theta_q}(z|c_i)|\mathcal{N}(0, 1))$
26:         **end for**
27:         Update the Critic with $\sum_{i=1}^{i=m} \mathcal{L}^i_{critic}$
28:         Update the context encoder with $\sum_{i=1}^{i=m}(\mathcal{L}^i_{critic} + \beta \mathcal{L}^i_{KL})$
29:         Update the path collector and the parameter generator with $\sum_{i=1}^{i=m} \mathcal{L}^i_{actor}$
30:     **end for**
31: **end for**

velocity running forward. For a training task, the target velocity is sampled uniformly from [0, 2.5]. For a test task, the target velocity is sampled uniformly from [2.5, 3.0]. The horizon length is set as 200. For the experiment, we sample 50 training tasks and 15 test tasks.

- Hopper-params: This environment is built by modifying the OpenAI gym Hopper, in which a two-dimensional one-legged robot should move forward. For each task, the dynamics parameters of the environment are randomly initialized. The horizon length is set as 200. For the experiment, we sample 40 training tasks and 10 test tasks.

- LunarLander-params: This environment is built by modifying the OpenAI gym Lunar Lander, in which a rocket should land on the landing pad. For each task, we randomly initialize the gravity. The horizon length is set as 1000. For the experiment, we sample 40 training tasks and 10 test tasks.

- InvDoublePend-params: This environment is built by modifying the OpenAI gym Inverted Double Pendulum. In this environment, there is a cart that can move linearly. A pole is fixed on it, and a second pole is fixed on the other end of the first pole. For each task, the dynamics parameters of the environment are randomly initialized. The horizon length is set as 200. For the experiment, we sample 40 training tasks and 10 test tasks.

Table 7: Measure of uncertainty for ESPL.

| Environment | Measure of uncertainty |
|---|---|
| CartPole | 0.4998 |
| MountainCar | 0.4998 |
| Pendulum | 0.4997 |
| InvDoublePend | 0.4995 |
| InvPendSwingup | 0.4999 |
| LunarLander | 0.4997 |
| Hopper | 0.4999 |
| BipedalWalker | 0.4998 |

- Cartpole-fl-ood: This environment is built by modifying the OpenAI gym Cart Pole[4]. In the environment, there is a cart that can move linearly with a pole fixed on it. For each task, we randomly initialize the force magnitude and the length of the pole. For a training task, the force magnitude and the length of the pole are sampled uniformly from $[7.5, 12.5]$ and $[0.3, 0.7]$. For a test task, the force magnitude and the length of the pole is sampled uniformly from $[5, 7.5] \cup [12.5, 15]$ and $[0.2, 0.3] \cup [0.7, 0.8]$. The horizon length is set as 200. For the experiment, we sample 40 training tasks and 10 test tasks.

For all the environments of single-task RL and meta-RL, we wrap the action interface with the *tanh* function to limit the range of the action.

## C  Implementation Details

### C.1  Single-task RL

During training, we employ two neural networks to approximate the Q function as the original SAC. The neural networks have two hidden layers with 256 hidden units. Besides, to limit the probabilities $p$ in the range of $[0, 1]$, we clip the probabilities after each update. After training, even if the mask is sampled from the Bernoulli distribution with probabilities $p$, the process of obtaining the symbolic policy is almost deterministic, because empirical observations show that almost all probabilities converge to 0 or 1 after training. We use a measure of uncertainty which is the average difference between probability $p$ and 0.5. When the measure approaches 0.5, the process of obtaining the symbolic policy is almost deterministic. We report the measure in Table 7. For the details of the training process, we give the pseudo-code in Algorithm 1.

### C.2  Meta-RL

In practice, we build up our off-policy learning framework on top of the soft actor-critic algorithm (SAC)[39] following PEARL. To construct a stochastic policy, we also employ a small neural network $F$ to output the standard deviation just like single-task RL. The neural network has two hidden layers with 64 hidden units. Note that this neural network is only used during training. During the evaluation, we only use the produced symbolic policy to infer the action. Besides, to limit the probabilities $p$ produced by $\Psi(z)$ in the range of $(0, 1)$, we employ the *sigmoid* function. We initialize the bias of the last layer in $\Psi(z)$ as 3.0. As $sigmoid(3.0) = 0.9526$, we will have most of the paths of the symbolic network active at the beginning of training and ensure that the input of the *sigmoid* function is not too large to prevent the gradient from disappearing during training. For the details of the training process, we give the pseudo-code in Algorithm 2.

---

[4]We use the same continuous version as single-task CartPole.

Table 8: Hyperparameters for the ESPL.

| Parameter | Value |
|---|---|
| optimizer | Adam[50] |
| number of samples per minibatch | 256 |
| scale of the reward | 1 |
| learning rate | $3 \cdot 10^{-4}$ |
| discount | 0.99 |
| sac target smoothing coefficient | 0.005 |
| target temperature | 0.2 |
| training steps per iteration | 1000 |
| scale of the penalty loss | 1 |

Table 9: Environment Specific Hyperparameters for ESPL.

| Environment | Scale of $\mathcal{L}_{select}$ | Target $l_0$ norm ratio | Schedule iterations |
|---|---|---|---|
| CartPole | 0.08 | 0.002 | 400 |
| MountainCar | 0.64 | 0.002 | 200 |
| Pendulum | 0.08 | 0.002 | 300 |
| InvDoublePend | 0.08 | 0.005 | 400 |
| InvPendSwingup | 0.1 | 0.005 | 400 |
| LunarLander | 0.08 | 0.005 | 600 |
| Hopper | 0.08 | 0.01 | 2000 |
| BipedalWalker | 0.64 | 0.01 | 1400 |

# D   Experiment details

## D.1   Single-task RL

In this section, we give the main hyperparameters of ESPL for single-task RL. We show the common hyperparameters of ESPL in Table 8. We also list the environment specific hyperparameters in Table 9. For the scale of $\mathcal{L}_{select}$, we choose the best one from $\{0.04, 0.08, 0.1, 0.16, 0.32, 0.64\}$. The target $l_0$ norm ratio is the ratio of the target $L_0$ norm of the mask at the end of training $l_e$ to the number of parameters of the symbolic network. We set the value according to the complexity of the task and do not tune the value. We show the learning curves of the average $L_0$ norm ratio of the mask for the sampled tasks in Figure 8. The $L_0$ norm ratio at the end of training is always higher than the target $l_0$ norm ratio. Thus, the $L_0$ norm ratio at the end of training is more affected by the scale of $\mathcal{L}_{select}$. The schedule iterations mean the number of iterations of temperature and target $L_0$ norm schedule. Then we illustrate the evaluation details about symbolic policies. DSP performs 3 independent training runs with different random seeds for each dimension of the action, selecting the best symbolic policy at the end of training. Thus, for environments with n-dimension action, they perform 3n training

Table 10: Hyperparameters for the CSP.

| Parameter | Value |
|---|---|
| optimizer | Adam[50] |
| number of samples per minibatch | 256 |
| scale of the reward | 5 |
| learning rate | $3 \cdot 10^{-4}$ |
| scale of the kl divergence loss | 1 |
| discount | 0.99 |
| sac target smoothing coefficient | 0.005 |
| target temperature | 0.2 |
| training steps per iteration | 2000 |
| scale of the penalty loss | 1 |

Table 11: Environment Specific Hyperparameters for CSP.

| Environment | Meta batchsize | Scale of $\mathcal{L}_{select}$ | Target $l_0$ norm ratio | Schedule iterations |
|---|---|---|---|---|
| Walker2d-params | 10 | 0.25 | 0.01 | 450 |
| Hopper-params | 10 | 0.25 | 0.01 | 300 |
| InvDoublePend-params | 10 | 2.0 | 0.01 | 150 |
| Cartpole-fl-ood | 10 | 0.25 | 0.002 | 25 |
| Lunarlander-g | 10 | 0.25 | 0.01 | 60 |
| Cheetah-vel-ood | 16 | 2.0 | 0.01 | 300 |

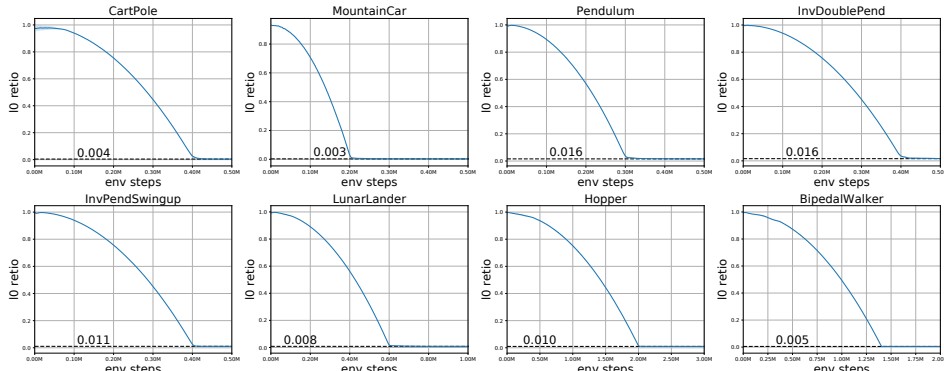

Figure 7: Learning curves of the average $L_0$ norm ratio of the mask for single-task RL.

runs and select the single best policy. The symbolic regression method repeats each experiment for a single dimension of action 10 times using different random seeds and selects the expression with the lowest mean-square error. For all the environments, the proposed ESPL performs 3 independent training runs and selects the single best policy.

## D.2 Meta-RL

In this section, we give the main hyperparameters of CSP for meta-RL. We show the common hyperparameters of CSP in Table 10. We also list the environment specific hyperparameters in Table 11. The meta batchsize is the number of sampled tasks per training step. We set it according to the number of training tasks. For the scale of $\mathcal{L}_{select}$, we choose the best one from $\{0.1, 0.15, 0.2, 0.25, 0.5, 1.0, 2.0\}$. The target $l_0$ norm ratio is the ratio of the target $L_0$ norm of the mask at the end of training $l_e$ to the number of parameters of the symbolic network. We set the value according to the complexity of the task and do not tune the value. We show the learning curves of the average $L_0$ norm ratio of the mask for the sampled tasks in Figure 8. The $L_0$ norm ratio at the end of training is always higher than the target $l_0$ norm ratio. Thus, the $L_0$ norm ratio at the end of training is more affected by the scale of $\mathcal{L}_{select}$. The schedule iterations mean the number of iterations of temperature and target $L_0$ norm schedule. We end the schedule near the end of training but for Cartpole-fl-ood in which the policy converges quickly, we reduce the number of schedule

Table 12: Average count of all selected paths and paths selected by at least ninety percent policies.

| Environment | Selected paths | Mostly selected paths |
|---|---|---|
| Walker2d-params | 76.42 | 70.3 |
| Hopper-params | 21.5 | 20.33 |
| InvDoublePend-params | 23.2 | 21.0 |
| Cartpole-fl-ood | 3.06 | 3.0 |
| Lunarlander-g | 5.2 | 5.0 |
| Cheetah-vel-ood | 27.04 | 18.5 |

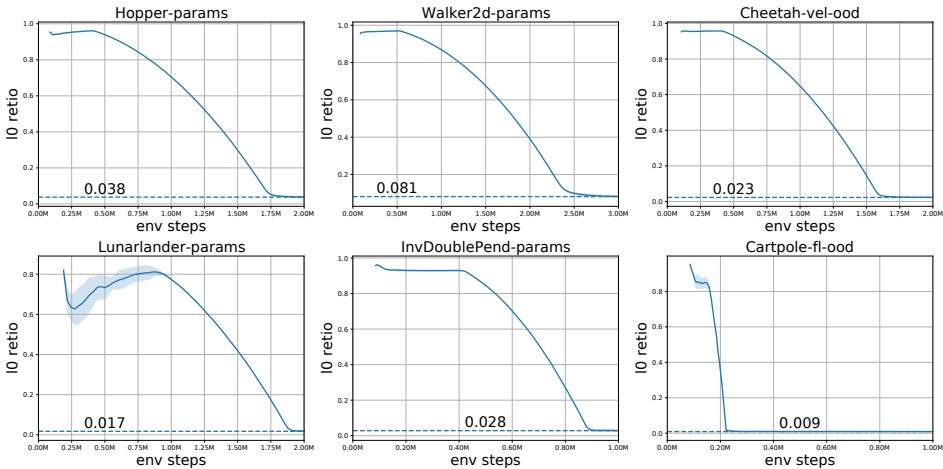

Figure 8: Learning curves of the average $L_0$ norm ratio of the mask for the sampled tasks in Meta-RL.

iterations. We run all the experiments with five random seeds and average the results to plot the learning curve.

### D.3 Platform and Device

The implementation of our ESPL and CSP is based on the pytorch[51]. Besides, we train the proposed CSP and ESPL with Nvidia V100 GPU. When evaluating the inference time, we use Intel(R) Xeon(R) Gold 5218R @ 2.10GHz CPU.

## E Extended Related Work

In this section, we provide an extended review of related work.

### E.1 Programmatic RL and program-guided RL

Programmatic RL represents policies as short programs to improve the interpretability of the policies. A series of works[18, 52, 19, 53, 54] obtain the programmatic policies by distilling them from neural network policies. VIPER[18] extracts neural network policies into a decision tree by imitation learning algorithm Q-DAGGER to get verifiable policies. LPP[52] also employ imitation learning but exploit the logical structure and use domain-specific language to avoid engineering each individual feature. NDPS[19] optimize the program by imitation learning from a neural network policy and employ an instance-specific syntactic model called sketches as guidance. PROPEL[53] views the unconstrained policy space as mixing neural and programmatic representations and uses mirror descent to optimize the policy. They evaluate their method in continuous control domains and show better results. LEAPS[54] learns the program embedding from given programs and uses CEM to search latent programs. MPPS[55] uses conditional variational auto-encoder and program synthesis to solve the partially observed problem and obtain the policy with MaxSAT. PRL[56] learns the programmatic policies by introducing a differentiable program derivation tree and searching the structure of the tree with gradient. Although programmatic RL methods also aim to obtain simple and interpretable policies, the ESPL proposed in this paper represents policies as symbolic or algebraic expressions which are different from programmatic RL. The different forms of symbolic expressions and programs lead to different difficulties in obtaining policies. This paper proposes a solution to the low data efficiency of the existing symbolic policy methods and extends it to meta-reinforcement learning. The scope of the application is also different. Some programmatic reinforcement learning[54, 55] interacts with the environment through DSL and is not applicable to continuous control tasks. In contrast, symbolic policy works often evaluate their methods in continuous control tasks, e.g. DSP. In addition, to further verify our method, we compared the proposed ESPL with some programmatic RL methods. Program-guided RL[57] trains the agent to

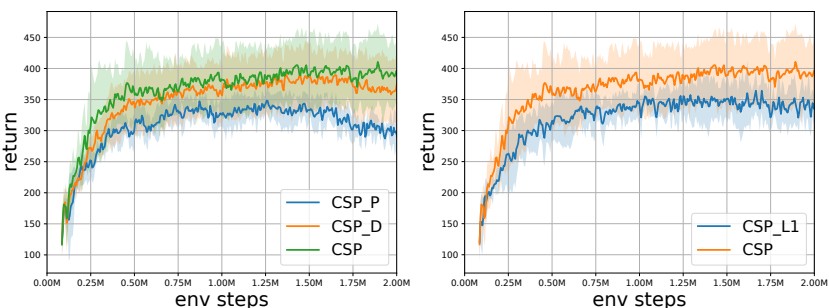

Figure 9: Ablation results of the symbolic network structure (left), the path selector(right). CSP_P means the plain structure and CSP_D means the densely connected structure. CSP_L1 means replacing the path selector with the $L_1$ norm minimization.

accomplish the tasks specified by a program and use a multitask policy to accomplish subtasks of the overall task. This approach is also interpretable but dependent on a given program, rather than learning an interpretable policy.

### E.2 Neural Architecture Search

Neural Architecture Search (NAS)[58, 59, 60] aims at automatically designing a neural network for a certain task and dataset and has been a promising approach to automate deep learning applications. DARTs[60] is one of the representative works of gradient-based neural architecture search [60, 61, 62]. These works formulate a super-network based on the continuous relaxation of the architecture representation, which allows efficient search of the architecture using gradient descent. Although we share the similar motivation to efficiently search by gradient descent and both have the process of selecting the appropriate substructure from a large structure, the algorithms are still quite different. These works use the softmax function to select one proper neural network operation from a set of candidate operations for each connection. In contrast, we learn to mask out redundant connections with the proposed path selector and only very small percentage of connections are selected (See Figure 7). In addition, they usually divide the computer vision dataset and alternate training structures and parameters, whereas we use a completely different training approach. Besides, these works aim to search for a good neural network structure for computer vision while we aim to find compact symbolic policies for reinforcement learning. We also involve meta-RL and develop contextual symbolic policies which can adapt to new tasks, while they usually need finetune or re-training for new tasks.

Building differentiable frameworks and using gradient-based methods to obtain solutions have indeed found applications in many fields, e.g. neural architecture search, and program induction [63, 64]. However, compared to ESPL, these works differ significantly in terms of (1) the constructed differentiable frameworks, (2) the process of obtaining solutions through gradients, and (3) the specific problem domains they target.

## F More Experiment Results

### F.1 Comparison with Programmatical RL

In the main body of this paper, we considered that the form of the policies will affect the final performance. Thus, except for neural network policies, we selected Regression and DSP, which also represent policies with symbolic expressions for comparison in order to ensure the fairness of comparison. We would like to compare the proposed ESPL with some programmatical RL methods to further verify our method in this section. We carry out experiments on The Open Racing Car Simulator (TORCS)[65] which provides a rich environment with input from up to 89 sensors.

We first follow the experiment settings in NDPS[19], where the controller tasks the input from 29 sensors and decides values for the acceleration, and steering. We show the lap time in seconds (lower is better) in Table 13.

Table 13: Comparison results in TORCS following experiment settings in NDPS.

|      | CG-SPEEDWAY-1 | AALBORG |
|------|---------------|---------|
| ESPL | **50.23**     | **107.3** |
| NDPS | 61.56         | 158.87  |

The results demonstrate that the symbolic policies produced by ESPL achieve higher performance. NDPS learns the programmatical policies from a neural network policy and the object mismatch problem may hurt the performance. We also provide the example video in `https://anonymous.4open.science/r/ESPL-8F5B`.

We then follow the experiment settings in PROPEL[53], where the controller tasks the input from 29 sensors and decides values for the acceleration, steering, and brake. We carry out experiments in three maps and show the lap time in seconds (lower is better) in Table 14. In this table, VIPER[18] is a method that extracts neural network policies into a decision tree by imitation learning. The PROIR means the PID controller that is also used as the initial policy in PROPEL. PROPELTREE means PROPEL using regression trees. PROPELPROG means PROPEL using a high-level language similar to NDPS. In these three maps, the symbolic policies produced by ESPL outperform other programmatical polcies and is comparable with PROPELTREE. The proposed ESPL train the symbolic policies from scratch without initial policies which are used in PROPEL. We also compare

Table 14: Comparison results in TORCS following experiment settings in PROPEL.

|               | G-Track | E-ROAD | AALBORG |
|---------------|---------|--------|---------|
| ESPL          | **77.31** | 79.67  | **107.94** |
| VIPER[18]     | 83.60   | 87.53  | 110.57  |
| PRIOR         | 312.92  | 322.59 | 244.19  |
| NDPS[19]      | 108.25  | 126.80 | 163.25  |
| PROPELPROG[53]| 93.67   | 119.17 | 147.28  |
| PROPELTREE[53]| 78.33   | **79.39** | 109.83 |

the proposed ESPL with PRL[56] on continuous control benchmarks presented in DSP in Table 15. PRL represents the policy with program derivation tree. They use affine policy which is the linear transformation as a domain-specific programming language (DSL) for Gym environments. The proposed ESPL achieve higher performance in most environments.

Table 15: Comparison with PRL on continuous control benchmarks presented in DSP.

|         | CartPole | MountainCar | Pendulum | InvDoublePend | InvPendSwingup | LunarLander | Hopper | BipedalWalker |
|---------|----------|-------------|----------|---------------|----------------|-------------|--------|---------------|
| ESPL    | 1000     | 94.02       | **-151.72** | **9359.9**  | **890.36**     | **283.56**  | **2442.48** | **309.43** |
| PRL[56] | 1000     | **95.07**   | -173.56  | 9357.93       | 889.47         | 265.17      | 2226.49 | 270.33       |

## F.2   Ablation Study for Meta-RL

In this section, we carry out experiments by ablating the features of CSP. We change the structure of the symbolic network and replaced the path selector with the $L_1$ norm minimization. we compare the learning curves of the test task rewards in Figure 9. The results show that the dense connections effectively improve the performance and we can facilitate the search for the proper symbolic form by arranging the operators to further improve the performance. Besides, the path selector is better at selecting the symbolic policy from the symbolic network than $L_1$ norm minimization.

## F.3   Human Study for Interpretability

We also carry out a human study to evaluate the interpretability of the symbolic policies generated by ESPL. The assessment of policy interpretability actually requires some understanding of the environments and what each state variable means. We invited ten researchers to rate the interpretability of policies, with a maximum of 20 minutes for each policy. On a five-point scale, we told them that

the interpretability could be judged based on whether they could see from the policy expression what the policies was based on to make decisions and the possible corresponding relationships between actions and states. A score of five indicated that the strategy was highly interpretable and could be designed by humans, while a score of zero indicated that it was completely uninterpretable, just like the neural network policies. We measured the average score of the interpretability of the policies. We give the interpretability scores for ESPL and DSP in Table 16.

Table 16: Comparasion of interpretability scores of ESPL and DSP.

|      | CartPole | MountainCar | Pendulum | InvDoublePend | InvPendSwingup | LunarLander | Hopper | BipedalWalker |
|------|----------|-------------|----------|---------------|----------------|-------------|--------|---------------|
| **ESPL** | 5 | 5 | 4.5 | 4.1 | 4.3 | 3.9 | 2.6 | 3.1 |
| **DSP**  | 5 | 5 | 4.5 | 5.0 | 4.2 | 4.0 | 3.1 | 3.2 |

Based on the results of the human study, we found that when the symbolic expressions were relatively short, there was a consensus that the expressions were highly interpretable. When the expression length became longer, the interpretability score decreased. It may be relatively difficult to understand symbolic expressions in a short time, but it is generally believed that interpretability is much higher than a black box, at least they can understand the action will be affected by which state, and the partial correlation. Besides, the symbolic policies produced by ESPL and DSP have comparable interpretability.

## F.4 Analysis of symbolic policies for Meta-RL

We then analyze the symbolic policies for different tasks produced by CSP. For each environment, we sample 10 tasks from the environment task distribution and obtain the corresponding symbolic policies with CSP. Then we analyze the selected paths of these policies which determine the forms of the symbolic expressions. Table 12 shows the results. We calculate the average count of selected paths per action dimension among the policies[5]. We find that this number varies across different environments. The symbolic expression can be extremely short for simple environments like Cartpole or relatively long for complex environments like Walker2D. We also calculate the average count of paths that are selected by more than ninety percent of the symbolic policies. In almost all environments, the mostly selected paths account for a high percentage of the selected paths, which indicates that the expressions of symbolic policies for different tasks of the same environment share similar forms.

## F.5 Theoretical Analysis for Cartpole

The theoretical analysis is as follows: The dynamic of cartpole system can be defined as:

$$\ddot{x} = \frac{8fa + 2m\sin\theta(4L\dot{\theta}^2 - 3g\cos\theta)}{8M - 3m\cos 2\theta + 5m}.$$

$$\ddot{\theta} = \frac{g\sin\theta - \frac{\cos\theta(fa + Lm\dot{\theta}^2\sin\theta)}{m+M}}{L(\frac{4}{3} - \frac{m\cos^2\theta}{m+M})}.$$

Where $f$ is the coefficient of force, $a$ represents the action, $m$ is the weight of the pole, $M$ is the weight of the cart, $L$ is the half-pole length, $\theta$ is the angle between the pole and the vertical direction, and $x$ denotes the horizontal coordinate of the cart.

Define $X = \begin{bmatrix} x \\ \dot{x} \\ \theta \\ \dot{\theta} \end{bmatrix}$, then $\dot{X} = \begin{bmatrix} \dot{x} \\ \frac{8fa + 2m\sin\theta(4L\dot{\theta}^2 - 3g\cos\theta)}{8M - 3m\cos 2\theta + 5m} \\ \dot{\theta} \\ \frac{g\sin\theta - \frac{\cos\theta(fa + Lm\dot{\theta}^2\sin\theta)}{m+M}}{L(\frac{4}{3} - \frac{m\cos^2\theta}{m+M})} \end{bmatrix}$.

---

[5]We only consider paths that contribute to the final expression. Besides, the count of the remaining paths may not equal to the count of operators in the final symbolic policy because some operators can be merged after simplifying symbolic expressions.

According to the Hartman-Grobman theorem, the local stability of this nonlinear system near its equilibrium point is equivalent to the linearized system near the equilibrium point. For cartpole system, the equilibrium point is $[x, \dot{x}, \theta, \dot{\theta}] = [0, 0, 0, 0]$

If $a = 0$, the system can be linearized as:

$$
\dot{X} = \begin{bmatrix} 0 & 1 & 0 & 0 \\ 0 & 0 & \frac{-6gm}{8M+2m} & 0 \\ 0 & 0 & 0 & 1 \\ 0 & 0 & \frac{g}{L(\frac{4}{3} - \frac{M}{m+M})} & 0 \end{bmatrix} \begin{bmatrix} x \\ \dot{x} \\ \theta \\ \dot{\theta} \end{bmatrix}.
$$

Calculate its eigenvalues:
$$[0, 0, 3.97114593, -3.97114593].$$

Due to the presence of positive eigenvalues, according to the Hartman-Grobman theorem, the system is unstable.

If $a = 17.17\theta + 1.2\dot{\theta}$, linearize the system near the equilibrium point:

$$
\dot{X} = \begin{bmatrix} 0 & 1 & 0 & 0 \\ 0 & 0 & \frac{137.36f - 6gm}{8M+2m} & \frac{9.6f}{8M+2m} \\ 0 & 0 & 0 & 1 \\ 0 & 0 & \frac{g - \frac{17.17f}{m+M}}{L(\frac{4}{3} - \frac{M}{m+M})} & \frac{-1.2f}{L(m+M)(\frac{4}{3} - \frac{M}{m+M})} \end{bmatrix} \begin{bmatrix} x \\ \dot{x} \\ \theta \\ \dot{\theta} \end{bmatrix}
$$

Calculate its eigenvalues:

$$[0 + 0.j, 0 + 0.j, -26.34 + 6.65014286j, -26.34 - 6.65014286j]$$

Since all the real parts of the eigenvalues are non-positive, according to the Hartman-Grobman theorem, the system is stable. Therefore, for the CartPole environment, the policies learned through ESPL can maintain the stability of the CartPole system.

