# OpenReview forum: "Efficient Symbolic Policy Learning with Differentiable Symbolic Expression"
_NeurIPS.cc/2023/Conference — NeurIPS 2023 poster_

### Official Review · Reviewer_9VjU · 2023-06-08

**Soundness:** 3 good
**Presentation:** 3 good
**Contribution:** 3 good
**Rating:** 5
**Confidence:** 5

**Summary:**

#  I have reviewed this draft once before.
This paper proposes a meta-RL method to generate explainable symbolic policies. ESPL contains a symbolic network in the search space and a path selector to find the compact symbolic policy.


**Strengths:**

I think there exists some novelty in ESPL because it contains symbolic + neural structures. Also, the symbolic policies induced by ESPL seem to be effective. The number of experiments is enough. The visualizations are cool.

**Weaknesses:**

Post rebuttal: I upgrade my score to a borderline accept because of the reasonable rebuttal.

Although I originally leaned to accept this paper, I think the suggestions from the previous conference are not incorporated, and the draft is not improved much, so this time, I have to vote for rejection.

The major concerns are:

(1) The proposed symbolic policy looks messy and it does not improve the interpretability much. Also, there could be many optimal policies.

(2) The CatPole and other approaches have guaranteed optimal policy. The theoratcal analysis is missing.

(3) The authors should try other discrete environments or complicated environments with rich semantics.

I feel sorry for this paper. It was very close to acceptance last time but the AC insisted that the novelties were not enough. I don't think a resubmission without major modifications can get this paper in.

**Questions:**

1. Standard errors should be shown.
2. Table3 is a waste of space.

**Limitations:**

Not discussed.

---

> ### Author Rebuttal · Authors · 2023-08-08
>
> Thanks a lot for your advice on further improving this paper. We would like to discuss them one by one. Any further discussion will be appreciated.
> >**Q1.**  I think the suggestions from the previous conference are not incorporated, and the draft is not improved much.
>
> Thank you again for the time and effort invested in the review process. We highly value the insightful feedback from the reviewers, which has been immensely beneficial to our work. We have diligently incorporated your suggestions provided during the previous conference review process. These revisions encompass additional discussions of related works (including NAS and Programmatical RL), experiments conducted in a more complex environment (TORCS), and comparisons with some Programmatical RL methods. Due to page limitations of NeurIPS, these modifications have been placed in the appendix. The specific revisions include:
> 1. Discussion of Programmatical RL and program-guided in Appendix E.1.
> 2. Discussion of Neural Architecture Search in Appendix E.2.
> 3. Comparison with Programmatical RL (NDPS,VIPER,PROPEL,PRL) in Appendix F.1.
> 4. Experiments in TORCS in Appendix F.1.
>
> >**Q2.** The proposed symbolic policy looks messy and it does not improve the interpretability much. Also, there could be many optimal policies.
>
> 1. The symbolic policies we obtained have a similar complexity compared to the previous works. In addition, the symbolic policies allow people to directly see what factors of the state are involved in the choice of action, as well as the rough relationship between the action and these factors. Besides, the superior interpretability to NN-policies has been discussed in previous works.
> 2. Despite the potential existence of optimal policies in certain environments, as far as we know, obtaining an analytical expression for such policies can be quite challenging or even infeasible. Even if attainable, it requires domain-specific knowledge and intricate computations. In contrast, our approach does not necessitate prior knowledge about the environment; it autonomously learns symbolic policies through interaction with the environment.
>
>
> >**Q3.** The CatPole and other approaches have guaranteed optimal policy. The theoretical analysis is missing.
>
> For complex nonlinear systems, we cannot obtain an analytical optimal policy. For simple systems like cartpole, since we can first know its dynamics and equilibrium points, one approach is to **approximate a linear system near the equilibrium point**, and use a linear quadratic regulator (LQR) for control. However, for other more complex environments, this becomes challenging to achieve. In contrast, our approach doesn't necessitate knowledge of the system's dynamics and can automatically learn symbolic policies.
>
> We provide the theoretical analysis of CartPole in the global rebuttal.
>
> >**Q4.** The authors should try other discrete environments or complicated environments with rich semantics.
>
> In this paper, we focus on tasks with continuous action space following previous works. For benchmark selection, we used the same benchmarks as in DSP to ensure a fair comparison, and we provided additional experimental results on other environments (TORCS) in the appendix.
>
>
> > **Q5.** It was very close to acceptance last time but the AC insisted that the novelties were not enough. I don't think a resubmission without major modifications can get this paper in.
>
> This is indeed frustrating, as the author was unable to see the discussion, we do not what happened in the final discussion. However, from the Paper Decision written by the AC, we did not see any content related to novelty. In fact, apart from summarizing the reviewers' opinions, the AC believes that there is a line of transformer-based SR methods after DSP that can also improve data efficiency should be compared. However, this is factually incorrect. Symbolic regression tasks and obtaining symbolic policies in reinforcement learning tasks are not the same thing. For symbolic regression tasks, one needs to regress expressions from the data generated by a certain symbolic expression, while symbolic policy learning aims to obtain symbolic policies that maximize cumulative rewards. Specifically, transformer-based SR requires ground-truth expressions during training, which might be suitable for supervised learning but is unreasonable for reinforcement learning. Unfortunately, we did not have the opportunity to rebuttal.
>
> Due to the page limitation, most of our modifications are in the appendix. We have added discussions on related works of NAS and Programmatical RL and conducted experiments on a more complex environment (TORCS). We have also included comparisons with some Programmatical RL methods.
>
>
> >**Q6.** Standard errors should be shown.
>
> In the table below, we give the mean and standard deviation of ESPL's episode rewards in the benchmark, but because previous works did not provide the standard deviation, we did not compare it in the table.
>
> |  | CartPole | MountainCar | Pendulum | InvDoublePend | InvPendSwingup | LunarLander | Hopper | BipedalWalker |
> |---|---|---|---|---|---|---|---|---|
> | ESPL | 1000$\pm$0.0 | 94.02$\pm$0.3 | -151.72$\pm$101.31 | 9359.9$\pm$1.54 | 890.36$\pm$38.80 | 283.56$\pm$20.35 | 2442.48$\pm$25.22 | 309.43$\pm$0.34 |
>
> >**Q7.** Table3 is a waste of space.
>
> Table 3 illustrates the number of episodes required for training. In contrast to DSP, ESPL utilizes significantly less data, showcasing its remarkable efficiency.

---

> > ### Comment · Reviewer_9VjU · 2023-08-10
> > **More detailed comments, and feedbacks to the authors' rebuttal.**
> >
> > Thanks to the authors for their rebuttals. Here are the extended comments and feedbacks on each point.
> > > 1) The proposed symbolic policy looks messy and it does not improve the interpretability much. Also, there could be many optimal policies. Although this paper claims to improve interpretability, however, the learned representations are still very noisy. I'm not sure whether this claim is true.
> >
> > I see that the authors added human experiments. However, honestly speaking I am not convinced by human experiments conducted in such a small temporal period.
> >
> > Besides, I'm more curious why the proposed approach is better than DSP in efficiency. I think both approaches use reinforcement learning, but the authors report that the proposed approach uses much fewer episodes to learn the policy. I'm not sure where the benefits come from.
> >
> > Do the authors compare DSP with an implementation of https://github.com/brendenpetersen/deep-symbolic-optimization? The authors say "These methods cannot be directly applied to learning symbolic policies." It is clear that their Github page contains applications in symbolic policy learning. I'm not sure whether the statement is true or not.
> >
> > > (2) Previous works show that the CartPole and other environments have guaranteed optimal policies [1]. I think the authors should compare their interpretability with [1]. More theoretical analysis of interpretability is encouraged, as previous papers show some proof of the quality of learned policy in simple environments. A smarter way is to not claim superiority in interpretability if the experiments fail to demonstrate this advantage.
> >
> > I think it is better to provide comparison demos (videos or other formats) instead of human evaluations. However, Table 1 only contains results from ESPL but does not show results from related baselines. It is noteworthy that the DSP paper contains several comparisons in Table 1.
> >
> > > (3) The authors should try other discrete environments or complicated environments with rich semantics. I believe the authors should try on MineCraft [2] or other higher-level environments. These semantic-rich environments require more interpretability. The logic is, we do not need a very interpretable policy in the low-level control tasks, such as PID. However, in higher-level planning tasks, we do need more interpretability.
> >
> > The authors respond that they focus on continuous action space. I think it is better to conduct experiments in discrete cases as well.
> >
> > [1] Verma, Abhinav, et al. "Programmatically interpretable reinforcement learning." International Conference on Machine Learning. PMLR, 2018. [2] Sun, Shao-Hua, Te-Lin Wu, and Joseph J. Lim. "Program guided agent." International Conference on Learning Representations. 2019.

---

> > > ### Comment · Reviewer_9VjU · 2023-08-11
> > > **More suggestions.**
> > >
> > > 1. If the authors still wish to claim superiority in interpretability, it is better to construct some hard cases that baseline models cannot interpret. For the existing environments, I think the DSP's Table 1 is already fairly simple.
> > >
> > > 2. The title needs to be changed to reveal the true novelty. The current title seems to claim they are the first to conduct symbolic policy learning via gradient descent. But apparently, this is not the truth.

---

> > > > ### Author Response · Authors · 2023-08-11
> > > > **Thank you very much for the suggestions!**
> > > >
> > > > Thanks very much for your further suggestion! We greatly value your suggestions and are willing to provide clarification and make modifications accordingly.
> > > >
> > > > 1. Actually, we **did not claim superiority in interpretability** in this paper. The motivation of this paper is to introduce a more efficient method for learning symbolic policies and eliminate the necessity for pre-trained policy models.
> > > >
> > > > 2. We are glad to modify the title. Our approach builds a fully differentiable framework and directly uses gradients to update expressions, whereas the previous approach (DSP) introduced RNNS as a proxy to generate symbolic expressions, and they used gradients to update the proxy instead of the symbolic expression itself. We understand your concern that the previous methodology also used gradients, so the title could be ambiguous.
> > > > How about changing the title to:
> > > > “Efficient Symbolic Policy Learning with Differentiable Symbolic Expression” or
> > > > "Efficient Symbolic Policy Learning by  Directly Updating the Expression with Gradient"

---

> > > > > ### Comment · Reviewer_9VjU · 2023-08-11
> > > > > **Thank you for your rebuttal.**
> > > > >
> > > > > I think “Efficient Symbolic Policy Learning with Differentiable Symbolic Expression” is a much better title. Also, the abstract and introduction need to be revised. It is better not to claim the interpretability as a contribution. Otherwise, you can tell the truth that the ESPL has a comparable outcome in terms of interpretability.
> > > > > Anyway, I'll increase my score because of the effort of the authors in rebuttal.

---

> > > > > > ### Author Response · Authors · 2023-08-11
> > > > > > **Thank You!**
> > > > > >
> > > > > > Thank you very much for your time and effort invested in the discussion! We appreciate your positive feedback and really gain a lot from the discussion. We are glad to reply and discuss if there are further concerns.

---

> > > ### Author Response · Authors · 2023-08-11
> > > **Thanks For Your Replay!**
> > >
> > > >**Q1.** I see that the authors added human experiments. However, honestly speaking I am not convinced by human experiments conducted in such a small temporal period. Besides, I'm more curious why the proposed approach is better than DSP in efficiency. I think both approaches use reinforcement learning, but the authors report that the proposed approach uses much fewer episodes to learn the policy. I'm not sure where the benefits come from.
> > >
> > > (1) The Human-study showed that the symbolic policies we learned can achieve close interpretability compared to DSP, and the experiment was conducted before the review comments were received.
> > > (2) It's important to note that our approach of **reinforcement learning is completely different from DSP**. DSP regards generating a symbolic policy as a sequential decision-making task and employs reinforcement learning to solve it, with the symbolic operation set as the action space. For the reinforcement learning process in DSP, a single reward (or in other word a single training example) requires interactions with the environment over multiple episodes (an episode may involve hundreds or thousands of interactions steps). In contrast, in our method, the learning of symbolic policies is gradient-based and **doesn't involve constructing any additional sequential decision-making process**. Each step of interaction of the symbolic policy and the environment serves as direct training example. Additionally, we have developed an off-policy training approach, which allows each training example to be used multiple times during training. Furthermore, when dealing with multi-dimensional actions, DSP needs to construct multiple sequential decision tasks. Simultaneously optimizing these tasks is challenging, leading them to use pre-trained neural network policies as anchor models and alternately optimizing each dimension of action. In contrast, we can optimize symbolic policies for all action dimensions simultaneously.
> > > Our approach is the first to require no prior environmental knowledge, no need for pre-trained policies, and offers a fully automated method for discovering symbolic policies.
> > >
> > > >**Q2.** Do the authors compare DSP with an implementation of https://github.com/brendenpetersen/deep-symbolic-optimization? The authors say "These methods cannot be directly applied to learning symbolic policies." It is clear that their Github page contains applications in symbolic policy learning. I'm not sure whether the statement is true or not.
> > >
> > > (1) Due to incomplete hyper-parameters of constant optimization in the DSP paper, reproducing the results is challenging. We utilized the same benchmark as DSP, allowing for a direct comparison with the results presented in the DSP paper.
> > > (2)  It is important to note that the portion of symbolic policy learning in the project found at https://github.com/brendenpetersen/deep-symbolic-optimization is exactly the official implementation of DSP. Symbolic policy learning and symbolic regression tasks are distinct. Symbolic regression aims to discover symbolic expressions that best fit a dataset using a fixed function. However, symbolic policy learning involves learning from interactions within an environment and has no access to a predefined dataset or symbolic expression. Consequently, applying symbolic regression methods directly to symbolic policy learning is not feasible. For instance, DSP required extensive modifications to apply the symbolic regression method [1] to symbolic policy learning.
> > >
> > > [1] Petersen, B. K., et al. Deep symbolic regression: Recovering mathematical expressions from data via risk-seeking policy gradients. Proc. of the International Conference on Learning Representations, 2021.

---

> > > > ### Author Response · Authors · 2023-08-11
> > > > **Thanks For Your Replay!**
> > > >
> > > > >**Q3.** Previous works show that the CartPole and other environments have guaranteed optimal policies [1]. I think the authors should compare their interpretability with [1]. More theoretical analysis of interpretability is encouraged, as previous papers show some proof of the quality of learned policy in simple environments. A smarter way is to not claim superiority in interpretability if the experiments fail to demonstrate this advantage. I think it is better to provide comparison demos (videos or other formats) instead of human evaluations. However, Table 1 only contains results from ESPL but does not show results from related baselines. It is noteworthy that the DSP paper contains several comparisons in Table 1.
> > > >
> > > > (1) As this paper is in our reference, we have reviewed the previous paper [1] and did not find indication of guaranteed optimal policies for Cartpole. The best policy presented in this paper also fails to consistently stabilize the system.
> > > >
> > > > (2) In our global rebuttal, we provided theoretical analysis of the effectiveness of the learned symbolic policy in the Cartpole environment.
> > > >
> > > > (3) Furthermore, note that the claims in this paper **do not encompass improved interpretability compared to prior symbolic policy works**. This paper aims to introduce a **more efficient method** for learning symbolic policies. The interpretability of symbolic policies is inherent to the policies themselves and is not a result of the symbolic learning method. Additionally, apart from interpretability, symbolic policies are simple and computationally lightweight.
> > > >
> > > > (4) Since we employed the exact same benchmark as DSP and DSP provided the learned symbolic policies, due to space constraints, we did not include a direct comparison with DSP's symbolic policies in the table. Furthermore, note that DSP did not provide a comparison with previous works in Table 1 of their paper, but instead contrasted with constant optimization of DSP. One reason is that earlier work only experimented on a small set of simple environments.
> > > >
> > > >
> > > > >**Q4.**  The authors should try other discrete environments or complicated environments with rich semantics. I believe the authors should try on MineCraft [2] or other higher-level environments. These semantic-rich environments require more interpretability. The logic is, we do not need a very interpretable policy in the low-level control tasks, such as PID. However, in higher-level planning tasks, we do need more interpretability.
> > > >
> > > > Discovering symbolic policies in discrete action space environments and discovering symbolic policies in continuous action space environments are two different research lines. They are also not compatible in terms of methodology. Previous works (such as DSP, PRIOR, PRL) that focus on continuous action spaces often do not consider environments like 2D Minecraft. Conversely, previous works that focus on grid-based environments typically do not evaluate their methods on continuous control tasks. These prior approaches frequently incorporate domain-specific programming languages (DSLs) designed by humans in their programs to handle environmental observation inputs and action outputs. However, the research line we are part of, which employs symbolic expressions as policies, is currently not compatible with these DSLs.
> > > > I acknowledge the significance of learning interpretable policies in higher-level environments. However, pursuing the research direction of learning symbolic policies in low-dimensional control tasks holds its own merit, encompassing both interpretability and efficiency during execution.
> > > >
> > > > Thanks again for your insightful feedback and for taking the time to communicate with us. We respectfully look forward to further discussion.

---

> > > > > ### Comment · Reviewer_9VjU · 2023-08-11
> > > > > **Novelty is not very clear.**
> > > > >
> > > > > Thanks for your additional response. It seems that the novelty against DSP is the construction of action space/reward design. However,
> > > > >
> > > > > (1) I don't think this can be called a completely different RL algorithm as the core parts (the model and the algorithm) are still the same. The difference lies in the action space and reward design. However, it is hard to motivate these changes.
> > > > >
> > > > > (2) If the proposed approach trains on multiple rounds of explored experiences, is this counted in the efficiency comparisons? Also, I think the idea of exploiting existing experiences is not novel, given the vast literature on sample efficient reinforcement learning.
> > > > >
> > > > > (3) The authors admit that they do not have superior interpretability. I agree with this point and I think the draft should be revised to remove Table 1 if the results are not better in terms of interpretability. These experiments only scatter the major goal of this paper.
> > > > >
> > > > > (4) How would the baseline perform if they use the same action space/reward design as yours? Is the model of ESPL better?
> > > > >
> > > > > (5) I don't know what the authors mean by "involve constructing any additional sequential decision-making process". In fact, what can be called "additional" is vague in the context.
> > > > >
> > > > > (6) Since the baseline already uses gradient-based RL approaches, the title of this draft seems to be too broad with an overclaim of novelty.

---

> > > > > > ### Author Response · Authors · 2023-08-11
> > > > > > **Clarify of Novelty**
> > > > > >
> > > > > > Thank you again for the time and effort invested in the review process.
> > > > > >
> > > > > > > (1)  I don't think this can be called a completely different RL algorithm as the core parts (the model and the algorithm) are still the same. The difference lies in the action space and reward design. However, it is hard to motivate these changes.
> > > > > >
> > > > > > **The core part (the model and the algorithm) is different.**
> > > > > > We attempt to explain differently:
> > > > > > DSP employs an RNN as the symbolic policy generator. Their algorithm consists of an inner RL loop and an outer RL loop (DSP Section 3, page 4).
> > > > > > Inner RL loop: The symbolic policy produces actions that interact with the environment. Due to the absence of well-defined gradients between actions and the RNN, they construct an additional RL loop.
> > > > > > Outer RL loop: DSP uses the RNN to gradually generate the symbolic policy (providing a symbolic operation, constant, or variable at each step). This RNN can be viewed as a policy.
> > > > > > The reward function for the outer RL loop is the average of trajectory rewards from multiple inner RL loops. As a result, DSP cannot efficiently utilize data generated from interactions with the environment.
> > > > > >
> > > > > > ESPL generates symbolic policies through a differentiable framework comprising a symbolic network and a path selector. We involve only a single RL loop, which involves interactions between the symbolic policy and the environment. As gradients are well-defined between the actions generated by the symbolic policy and the symbolic network/path selector, we can directly update the symbolic policy without constructing an additional RL loop. This allows us to efficiently leverage data from interactions with the environment, leading to higher efficiency.
> > > > > >
> > > > > >
> > > > > > >(2) If the proposed approach trains on multiple rounds of explored experiences, is this counted in the efficiency comparisons? Also, I think the idea of exploiting existing experiences is not novel, given the vast literature on sample efficient reinforcement learning.
> > > > > >
> > > > > > Our method also requires multiple rounds of interaction with the environment, which are stored in the replay buffer and can be used multiple times. Training procedures are given in the Appendix.
> > > > > > In addition, it is because we build a differentiable symbolic policy framework that we can exploit existing experiences.
> > > > > >
> > > > > > > (3) The authors admit that they do not have superior interpretability. I agree with this point and I think the draft should be revised to remove Table 1 if the results are not better in terms of interpretability. These experiments only scatter the major goal of this paper.
> > > > > >
> > > > > > We Never claim superiority in interpretability in this paper. The purpose of Table 1 is to prove that our method is able to learn symbolic policies.
> > > > > >
> > > > > > >(4) How would the baseline perform if they use the same action space/reward design as yours? Is the model of ESPL better?
> > > > > >
> > > > > > They **can not** use the same action space/reward design. It is the proposed differentiable framework that enables the action space/reward design.
> > > > > >
> > > > > > > (5) I don't know what the authors mean by "involve constructing any additional sequential decision-making process". In fact, what can be called "additional" is vague in the context.
> > > > > >
> > > > > > In fact, symbolic policies or, more broadly, reinforcement learning is designed to solve sequential decision problems. "Cartpole" and "Hopper" are sequential decision problems. DSP also models the process of producing symbolic policies as an additional sequential decision problem. We don't need to introduce such a procedure.
> > > > > >
> > > > > > (6) Since the baseline already uses gradient-based RL approaches, the title of this draft seems to be too broad with an overclaim of novelty.
> > > > > >
> > > > > >  Our approach builds a fully differentiable framework and directly uses gradients to update expressions, whereas the previous approach (DSP) introduced RNNS as a proxy to generate symbolic expressions, and they used gradients to update the proxy instead of the symbolic expression itself.  We are willing to modify the title to “Efficient Symbolic Policy Learning with Differentiable Symbolic Expression” or
> > > > > > "Efficient Symbolic Policy Learning by Directly Updating the Expression with Gradient"

---

### Official Review · Reviewer_PhEA · 2023-07-04

**Soundness:** 2 fair
**Presentation:** 3 good
**Contribution:** 2 fair
**Rating:** 6
**Confidence:** 3

**Summary:**

This paper proposes to apply differentiable symbolic regression for policy learning. It shows promising results in multiple RL environments (including good average performance and learned interpretable symbolic policies).

**Strengths:**

This paper shows promising results of differentiable symbolic regression in eight RL environments. The model design is intuitive and is shown to achieve good average performance, learn interpretable symbolic policies, and be less likely to overfit for OOD generalization in the meta-RL setting. The model is also more sample efficient than a previous neural symbolic regression model for policy learning.

**Weaknesses:**

Nevertheless, the model is less novel as similar techniques have been explored in domains such as differentiable interpreters (e.g., Terpret [1] or DiffForth [2]) for program synthesis and Dart [3] for neural architecture search/pruning.

There are also concerns regarding the experiments:
* Symbolic methods are reported to perform better than neural ones even for single-task RL settings, which is counter-intuitive. There should not be overfitting problems in the naive setting, as far as I understand. Why would symbolic methods outperform neural ones?
  - I find this in appendix, lines 123-124, which seems relevant but could be unfair:
   > For all the environments, the proposed ESPL performs 3 independent training runs and selects the single best policy.
  - I also find that they may report averaged performance over multiple runs, which may not be necessary or standard for the single-task RL setting.
* Overall, the descriptions regarding evaluation procedures are vague, e.g.,
  - it's unclear whether the final policies evaluated in Table 2 are symbolic or not; and
  - how to select policies in Table 1;


[1] Gaunt, Alexander L., et al. "Terpret: A probabilistic programming language for program induction." arXiv preprint arXiv:1608.04428 (2016).

[2] Bošnjak, Matko, et al. "Programming with a differentiable forth interpreter." International conference on machine learning. PMLR, 2017.

[3] Liu, Hanxiao, Karen Simonyan, and Yiming Yang. "Darts: Differentiable architecture search." arXiv preprint arXiv:1806.09055 (2018).

**Questions:**

* Why would the symbolic methods outperform the neural ones in the single-task RL settings?
* Are there results in more complex/practical environments, e.g., with larger input and output dimensions or longer expressions?

---

> ### Author Rebuttal · Authors · 2023-08-08
>
> Thanks for your detailed review. We are glad to discuss your concerns one by one. Any further discussion will be appreciated.
> >**Q1.** Nevertheless, the model is less novel as similar techniques have been explored in domains such as differentiable interpreters (e.g., Terpret [1] or DiffForth [2]) for program synthesis and Dart [3] for neural architecture search/pruning.
>
> A1. The proposed framework is novel and is different from previous works. Building differentiable frameworks and using gradient-based methods to obtain solutions have indeed found applications in many fields. However, compared to ESPL, these works differ significantly in terms of (1) the constructed differentiable frameworks, (2) the process of obtaining solutions through gradients, and (3) the specific problem domains they target. For example, Darts formulate a super-network based on the continuous relaxation of the neural architecture representation, while we employ a densely connected symbolic network. Darts uses the softmax function to select one proper neural network operation from a set of candidate operations for each connection. In contrast, we learn to mask out redundant connections with the proposed path selector and only a very small percentage of connections are selected. Darts aims to search for a good neural network structure for computer vision while we aim to find compact symbolic policies for reinforcement learning. In Appendix E2, we have compared our proposed method with Neural Architecture Search approaches. We will add these works into the reference.
>
> >**Q2.** Symbolic methods are reported to perform better than neural ones even for single-task RL settings, which is counter-intuitive. There should not be overfitting problems in the naive setting, as far as I understand. Why would symbolic methods outperform neural ones? I find this in appendix, lines 123-124, which seems relevant but could be unfair:'For all the environments, the proposed ESPL performs 3 independent training runs and selects the single best policy.' I also find that they may report averaged performance over multiple runs, which may not be necessary or standard for the single-task RL setting.
>
> A2. In fact, for a fair comparison, we adopted the **exact same testing procedures** as in DSP[1] which is the SOTA of symbolic policy learning and the most important baseline, including selecting from 3 independent training runs and averaging over multiple runs. It is worth noting that neural network policies are not necessarily superior to symbolic policies, and this is also evident from the experimental results in DSP[1]. One possible reason is that symbolic policies can represent complex relationships through combinations of different symbolic operations. When the symbolic expression is already sufficient to represent the policy, optimizing fewer parameters may be simpler and more effective.
>
> [1]Landajuela, Mikel, et al. "Discovering symbolic policies with deep reinforcement learning." International Conference on Machine Learning. PMLR, 2021.
>
> >**Q3.** Overall, the descriptions regarding evaluation procedures are vague, e.g., it's unclear whether the final policies evaluated in Table 2 are symbolic or not; and how to select policies in Table 1;
>
> A3. The average episode rewards of ESPL presented in Table 2 are obtained by evaluating the final obtained **symbolic policies**. Additionally, the symbolic policy given in Table 1 is the one used for evaluation. We followed the same method as in DSP to select the symbolic policy, performing independent training runs and selecting the single best policy (see Appendix lines 123-124). We will add a description in revision to make it clearer.
>
>
> >**Q4.** Are there results in more complex/practical environments, e.g., with larger input and output dimensions or longer expressions?
>
> A4. In fact, for a fair comparison, we used the exact same benchmark as in DSP (line 240-241). This benchmark includes classic control tasks such as CartPole and complex tasks like BipedalWalker, whereas earlier symbolic policy works were only evaluated on simpler tasks, such as CartPole or 1-DOF robot arm. Additionally, we included some additional experimental results (more environments and comparisons) in Appendix F.

---

> > ### Comment · Reviewer_PhEA · 2023-08-10
> >
> > Thanks to the authors for the rebuttal. I have read the rebuttal but, unfortunately, cannot find convincing results to adjust my rating. In more detail,
> > * regarding results and evaluation, following one previous work may not be sufficient to justify the evaluation procedure. It would be better if the authors could justify more from the perspective of tasks' needs, etc.
> > * regarding novelty, I was aware of the differences as the authors listed in the rebuttal, but do not think they are significant. As stated in my review, I consider this work as old methods/principles applied in new domains (therefore *less* novel).
> >
> > Overall, with these concerns in mind, I will keep my rating as 5 (borderline accept) for now.

---

> > > ### Author Response · Authors · 2023-08-13
> > > **Thanks for your reply!**
> > >
> > >  Thanks for your insightful feedback and for taking the time to communicate with us.
> > >
> > >  > **Q1.** regarding results and evaluation, following one previous work may not be sufficient to justify the evaluation procedure. It would be better if the authors could justify more from the perspective of tasks' needs, etc.
> > >
> > > 1. The DSP is, in fact, the SOTA in the symbolic policy domain and is the most essential baseline currently. To validate the effectiveness of our method, we employed benchmarks of DSP and compared our results against the scores presented in the DSP paper. To ensure fairness in comparison, we have to utilize the same testing methodology as employed by DSP. Furthermore, in the meta-RL experiments, for a fair comparison, we did not select symbolic policies. We adopted the same testing methodology as PEARL.
> > > 2. Due to the stochastic nature of environment initialization, averaging over multiple runs better reflects the algorithm's performance. Moreover, in the field of symbolic or programmatical policy learning, select symbolic policies from multiple independent runs is a common practice[1][2][3][4], aiding in preventing the search from being trapped in local optima.
> > >
> > > [1]  Jiˇrí Kubalík, Eduard Alibekov, and Robert Babuška. Optimal control via reinforcement learning with symbolic policy approximation. IFAC-PapersOnLine, 50(1):4162–4167, 2017. 20th IFAC World Congress.
> > > [2]  Verma, Abhinav, et al. "Programmatically interpretable reinforcement learning." International Conference on Machine Learning. PMLR, 2018.
> > > [3] Bastani, Osbert, Yewen Pu, and Armando Solar-Lezama. "Verifiable reinforcement learning via policy extraction." Advances in neural information processing systems 31 (2018).
> > > [4] Landajuela, Mikel, et al. "Discovering symbolic policies with deep reinforcement learning." International Conference on Machine Learning. PMLR, 2021.
> > >
> > > > **Q2.** regarding novelty, I was aware of the differences as the authors listed in the rebuttal, but do not think they are significant. As stated in my review, I consider this work as old methods/principles applied in new domains (therefore less novel).
> > >
> > > A2. Building differentiable frameworks and using gradient-based methods to obtain solutions is a conceptual approach rather than a one-size-fits-all solution.  There is quite a bit of works in the NAS field alone using this approach. But it does take much effort to design algorithms for specific problems using this approach. In this paper, our novelty is reflected in the design of symbolic network structure, probabilistic path selector and corresponding training algorithm to build the first-ever differentiable symbolic policy learning framework for efficient policy learning.
> > >
> > > Thank you again for your insightful feedback and constructive comments.

---

> > > > ### Comment · Reviewer_PhEA · 2023-08-13
> > > >
> > > > I understand the authors' difficulties in experimental designs and appreciate the efforts in model designs. Nevertheless, I was aware of them while reviewing the paper. Therefore, I will still keep my rating as 5 (borderline accept).
> > > >
> > > > For what it's worth, in the single-task RL setting, it would be preferable and more practical to analyze the best performance among all runs or the sample efficiency to obtain good performances, from my perspective. It would thus be better to include this setting in the experiments (along with the original setting to compare with DSP). ESPL does not need to outperform DRL in numbers as it has other advantages. \
> > > > The authors had better cite the related works thoroughly as well to avoid the risk of overclaiming novelty.

---

> > > > > ### Author Response · Authors · 2023-08-16
> > > > > **Thanks for your advice!**
> > > > >
> > > > > Based on your suggestions, we conduct new experiments. For all DRL algorithms, we further tune the hyperparameters from rl-baselines-zoo and conducted three independent training runs. For each run, we select the best-performing policies across all time steps. Then we select the best performance of the three training runs as the final results. The results are shown in the following table. TD3 and SAC outperform other algorithms. For symbolic policy methods, the proposed ESPL achieves superior performance compared to DSP. ESPL is also comparable with the better-performing algorithms of DRL.
> > > > >
> > > > > We discuss the relevant work of NAS in the appendix, and we will thoroughly cite other related works in the revision.
> > > > >
> > > > > | Environment | DDPG | TRPO | A2C | PPO | ACKTR | SAC | TD3 | Regression | DSP | ESPL |
> > > > > |---|---|---|---|---|---|---|---|---|---|---|
> > > > > | Cartpole | 1000 | 1000 | 1000 | 1000 | 1000 | 1000 | 1000 | 211.82 | 1000 | 1000 |
> > > > > | Mountaincat | 95.36 | 93.6 | 93.97 | 93.76 | 93.79 | 94.68 | 93.87 | 95.16 | 99.11 | 94.02 |
> > > > > | Pendulum | -155.6 | -145.49 | -157.59 | -160.14 | -201.57 | -154.82 | -155.06 | -1206.9 | -155.4 | -151.72 |
> > > > > | InvDoublePend | 9347.1 | 9188.43 | 9359.81 | 9356.59 | 9359.06 | 9359.92 | 9359.25 | 637.2 | 9149.9 | 9359.9 |
> > > > > | InvPendSwingup | 891.48 | 892.9 | 254.71 | 890.1 | 890.11 | 891.32 | 892.25 | -19.21 | 891.9 | 890.36 |
> > > > > | LunarLander | 266.05 | 265.26 | 238.51 | 269.65 | 271.53 | 276.92 | 272.13 | 56.08 | 261.36 | 283.56 |
> > > > > | Hopper | 1678.84 | 2593.56 | 2104.98 | 2586.56 | 2583.88 | 2613.16 | 2743.9 | 47.35 | 2122.4 | 2442.48 |
> > > > > | BipedalWalker | 209.42 | 312.14 | 291.79 | 287.43 | 309.57 | 308.31 | 314.24 | -110.77 | 311.78 | 309.43 |
> > > > > | Worst Rank | 9 | 10 | 9 | 9 | 9 | **6** | 7 | 10 | 9 | **6** |
> > > > > | Average Rank | 5.5 | 4.125 | 6.25 | 6.125 | 5.375 | 3 | **2.875** | 9.125 | 4.625 | 3.5 |
> > > > >
> > > > > Thanks again for your time and effort invested in the review!

---

> > > > > > ### Comment · Reviewer_PhEA · 2023-08-16
> > > > > >
> > > > > > I appreciate the authors' efforts in rebuttal and thus decided to adjust my rating to 6 (weak accept). If accepted, I hope the authors will revise the paper as stated, including (1) a clearer and more detailed explanation of the evaluation procedure; (2) "fairer" experimental settings (even in the meta-RL setting, if doable, for the benefit of the field); and (3) a thorough discussion about the novelty.

---

> > > > > > > ### Author Response · Authors · 2023-08-17
> > > > > > > **Thank you!**
> > > > > > >
> > > > > > > We sincerely appreciate your constructive feedback and the effort you have invested during the review and discussion phases.
> > > > > > >
> > > > > > > We will carefully revise the paper according to your suggestions. Regarding meta-RL, since the previous symbolic policy methods do not support meta-RL, we compare ESPL with neural network-based approaches and adopt the same evaluation procedure of these methods which is also commonly used in the meta-RL field to ensure fairness in comparison.
> > > > > > > We will provide a clearer and more detailed explanation of the evaluation procedure for both single RL and meta RL, incorporate the experiment results and theoretical analysis during rebuttal and discussion, discuss the novelty, and make sure to cite other related works in the revision.

---

### Official Review · Reviewer_uiuA · 2023-07-06

**Soundness:** 3 good
**Presentation:** 2 fair
**Contribution:** 2 fair
**Rating:** 5
**Confidence:** 4

**Summary:**

The paper "Efficient Symbolic Polmicy Learning via Gradient Descent" proposes a new neural symbolic architecture for agents learned via reinforcement learning. Authors propose an architecture with an alternance of symbolic and linear layers. To finally obtain simple expressions, a probabilistic mask is learned altogether with parameters, enabling the production of sparse archiectures resulting in small expressions. The approach is experimented on classical RL settings, and also on meta-rl ones.

**Strengths:**

- Very interesting architecture
- Proposals look innovative
- Clever way to obtain sparsity
- Convincing experiments about efficiency

**Weaknesses:**

- I cannot understand, if the architecture is new, why authors did not experiment it on classical supervised learning (eg, on classification tasks) before such much more reinforcement learning setting. Is the architecture only new in the field of RL ? Is the main contribution of the paper the archiecture or its application to RL and meta-RL ?
- I feel that lacks important details about the used archiectures (for instances q is not defined, nothing is said about the choice of symbolic operators, nor about their numbers n and m, etc.)
- Not much results about explanability, which is one of the main goal of the proposal

**Questions:**

- Is the main contribution of the paper the archiecture or its application to RL and meta-RL ? what about applying it on supervised learning tasks ?
- How is q defined ? is it a deterministic output from a lstm or a gaussian from a transformer or ...? How is it trained ?
- How are chosen the sequence of symbolic operators used in the architecture and what are they in the experiments ? how many (please specify m and n)
- From my understanding, since we have m+2n inputs in the symbolic layer and only m+n outputs, the architecture has a with that reduces with layers. is it true ?
- What is the impact of lmin in (5) ? Do you have experiments to analyze it ?
- The metaRL objective given in the end of the first paragraph of page 6 is difficult to understand: the context is both sampled from the policy and used by the policy. Could you give clearer insights about this strange cycling definition ? Also, I would have expect that there is for each new task an exploration phase that would not been considered in the quantity to maximize : for k episodes no rewards, only collecting experience, such that we are good for the k+1th one in expectation depending on contexts from the k first episodes. Wouldn't it be better to present it like this ?





**Limitations:**

.

---

> ### Author Rebuttal · Authors · 2023-08-08
>
> Thanks for the thoughtful comments. We would like to clarify the concerns as follows:
>
> >**Q1.** I cannot understand, if the architecture is new ...
>
> A1. (1) This paper aims to propose an efficient and effective method for learning symbolic policies. Symbolic policies are designed for sequential decision tasks, which are typically addressed using reinforcement learning methods rather than supervised learning.
> (2) This approach is innovative and not a direct application of existing method of reinforcement learning.
> However, it cannot be directly applied to supervised learning tasks. Supervised learning tasks (e.g. image classification, machine translation) have very different input and output forms compared to sequential decision-making tasks. As for the symbolic network, we drew inspiration from prior works (line 116) in the field of symbolic regression. However, our symbolic network architecture, constraints on symbolic operations, training process, and the process of obtaining compact expressions are distinct. These methods cannot be directly applied to learning symbolic policies.
> (3) The contributions of this paper are three-fold. First, we introduce a novel gradient-based symbolic policy learning algorithm named ESPL that efficiently learns symbolic policies from scratch. The novelty is reflected in new symbolic network architecture, regularized operators, probabilistic path selectors, new loss functions, and training procedures, among others. Next, with ESPL, we develop the contextual symbolic policy for meta-RL, enabling the generation of symbolic policies for unseen tasks. This is a capability not achieved by any previous symbolic policy learning methods. Finally, we summarize our empirical results, which demonstrate the advantages of ESPL in both single-task RL and meta-RL scenarios.
>
> >**Q2.** I feel that lacks important details about the used ...
>
> A2. The Q-function (also known as the state-action-value function) is a crucial function in reinforcement learning, serving as a means to assess the expected return of different actions taken in a certain state. The Q-function is also referred to as the critic, and we update its parameters using the SAC algorithm, with pseudocode provided in Appendix C. The selection of symbolic operators is detailed in Section 3. To ensure a fair comparison, we adopt the same set of symbolic operations as in DSP. The architecture of the symbolic network used in experiments follows the structure shown on the right side of Figure 1 (lines 154-155). The values of m and n can also be derived from the network structure. For instance, if the next-layer symbolic operations are $[exp, log, mul, div]$, then m=2 and n=2, where m represents the number of unary functions and n represents the number of binary functions (line 123).
>
>
> >**Q3.** Not much results about explainability ...
>
> A3. (1) We give examples of interpretable analysis in Section 5.4 and report the symbolic policy lessons learned in Table 1. The symbolic policies allow people to directly see what factors of the state are involved in the choice of action, as well as the rough relationship between the action and these factors.
> (2) It has been a consensus in previous works that using symbolic expression forms of policies has better interpretability [1][2][3]. In this paper, our main motivation is to design an **efficient** symbolic policy learning method, which greatly reduces the number of interaction episodes required for learning symbolic policies and extends the application of symbolic policies to the field of meta-RL.
> (3) To further measure the interpretability of the learned symbolic policies, we present a human-study in the global rebuttal.
>
> [1] Discovering symbolic policies with deep reinforcement learning. ICML 2021
> [2] Optimal control via reinforcement learning with symbolic policy approximation.
> [3] Interpretable policies for reinforcement learning by genetic programming.
>
> >**Q4.** Is the main contribution of the paper the architecture ...
>
> A4. We outlined our contributions in A1. It is also possible to apply symbolic approaches to solve supervised learning problems (such as image classification and machine translation), but our symbolic policy learning method cannot be directly employed for this purpose.
>
> >**Q5.** How is q defined ? ...
>
> A5. For the definition and training methodology of Q, please refer to A2. In the experiments, the network structure for Q is a multi-layer perceptron (MLP) with a hidden layer size of [256, 256].
>
> >**Q6.** How are chosen the sequence of symbolic ...
>
> A6. Please refer to A2.
>
> >**Q7.** From my understanding, since we have m+2n inputs in the symbolic layer and only m+n outputs, the architecture has a width that reduces with layers. is it true ?
>
> A7. The architecture dose not have a width that reduces with layers. There is also a fully connected layer between the symbolic layer (line 124-125), and the output dimension of the fully connected layer is always the input dimension (m+2n) required for symbolic operations in the next layer, so the width does not gradually decrease. The width depends on the symbolic operators in the next layer.
>
> >**Q8.** What is the impact of lmin in (5) ? Do you have experiments to analyze it ?
>
> A8. To constrain the minimal complexity of symbolic policies, we introduce the minimum L0 norm denoted as $l_{min}$ (line 187). We gradually reduce $l_{min}$ from the count of the original parameters w to a specified target value using a parabolic function (lines 195-196). As training progresses, this approach leads to a gradual reduction in the complexity of symbolic policies, resulting in a compact expression. The learning curve for the complexity of symbolic policies is presented in Appendix Figure 3. We intuitively set the value for the target $l_{min}$ and do not tune it.
>
> >**Q9.** The metaRL objective given in the end of the first paragraph of page 6 is difficult to understand
>
> A9. We clarify the metaRL in the global rebuttal.

---

> > ### Comment · Reviewer_uiuA · 2023-08-11
> >
> > Thanks to authors for their insightful answers.
> >
> > However I still do not understand why it could not be applied to symbolic regression or classification tasks: why must be the problem be a sequential decision one to apply your method ?
> >
> > For me this looks similar to researches on neural network architectures (such things like [1] for instance), but with special symbolic operators at each layer. These can be applied to supervised classification tasks...
> >
> > I slightly improve my score (but also still think that presentation should be somehow reworked to better detail each component and give better intuition about the proposal. For instance more discussion about how are symbolic operators arranged as illustrated in fig 1 would be welcome, maybe giving ablations regarding this aspect of the contribution)
> >
> > [1] Tom Veniat, Ludovic Denoyer:
> > Learning Time/Memory-Efficient Deep Architectures With Budgeted Super Networks. CVPR 2018: 3492-3500

---

> > > ### Author Response · Authors · 2023-08-13
> > > **Thanks for your reply!**
> > >
> > > Thank you very much for your positive feedback!
> > >
> > > > **Q1.** However I still do not understand why it could not be applied to symbolic regression or classification tasks: why must be the problem be a sequential decision one to apply your method
> > >
> > >
> > > A1. 1. Our framework is designed to efficiently learn symbolic policies, involving reinforcement learning processes and corresponding meta-RL designs tailored for sequential decision-making processes.
> > > 2. When talking about symbolic networks and probabilistic path selectors alone, they could be applicable to supervised learning, but it introduces additional challenges. For instance, classification tasks often entail high-dimensional inputs, whereas symbolic expressions typically require lower-dimensional inputs. This might necessitate integration with certain feature extraction methods, such as beta-VAE, and Slot Attention. Indeed, this presents an intriguing direction for future research.
> > >
> > > > **Q2.** For me this looks similar to researches on neural network architectures (such things like [1] for instance), but with special symbolic operators at each layer. These can be applied to supervised classification tasks...
> > >
> > > A2. Our approach is related to some network architecture search works. While we search for symbolic policies from the symbolic network, a series of network architecture search works construct hypernetworks and search for high-performing or computationally efficient network structures from these hypernetworks. However, our approach differs in terms of network construction, search algorithms, and the target task. We have provided a comparison with related works of NAS in Appendix E2.
> > >
> > > > **Q3.** More discussion about how are symbolic operators arranged as illustrated in fig 1 would be welcome, maybe giving ablations regarding this aspect of the contribution.
> > >
> > > A3. As described in lines 155-157, we heuristically involve more multiplication and division operators at shallow layers to provide more choice of input processed by simple operators for complex operations such as sines and cosines. We also provide an ablation study in Section 5.5. The results are presented in Table 5. Both the path selector and dense connections play a crucial role in the performance of learned symbolic policies. When the path selector is replaced with L1 or dense connections are removed, performance significantly deteriorates in certain environments. When employing dense connections and the path selector, but without symbolic arrangement (as shown in the middle structure of Figure 1), the approach can perform well in all environments. However, our heuristic symbolic arrangement further enhances the algorithm's performance. We also provide the ablation study results for meta-RL in Appendix F.
> > >  We will revise the representation for better clarity.
> > >
> > >  Thanks again for your time and effort invested in the review!

---

### Official Review · Reviewer_iXeo · 2023-07-07

**Soundness:** 3 good
**Presentation:** 4 excellent
**Contribution:** 4 excellent
**Rating:** 7
**Confidence:** 4

**Summary:**

The paper proposes ESPL, a method for learning symbolic policies in environments with low-dimensional state spaces. ESPL uses a densely connected neural network structure (like DenseNet), where the activations in each layer are replaced with a hand-picked set of functions, such as multiplication, division, log and and exp. Second, ESPL uses the Gumbel-Softmax trick to learn a masking function with a minimal $L_0$ norm. This function ensures that the method learns compact symbolic policies by masking out redundant pathways in the neural network. ESPL is shown to be 100 to 1000 times more sample-efficient than a prior symbolic learning approach in single-task RL. The authors also propose a version that can be conditioned on task context for meta-RL.

**Strengths:**

1. The proposed method is well motivated and clearly explained. $L_0$ regularization of the parameters of Bernoulli random variables is an elegant approach to learning compact symbolic expressions.
2. In single-task RL, ESPL outperforms a prior symbolic learning approach $DSP^0$ with between 100 and 1000 fewer training samples.
3. A task-conditioned version of ESPL called CSP outperforms prior (non-symbolic) approaches in a meta-learning setup. The authors point out that the symbolic policies are much faster to run than the baseline neural net policies.
4. Source code is included.

**Weaknesses:**

1. The discovered symbolic policies in Table 1 seem to be somewhat more complex than the policies from $DSP^0$. It could be useful to additionally measure the complexity of the discovered policies for the various methods (if such measure exists).
2. It is not very clear how much effort went into hand-picking the activation functions in each layer of the network. Is it necessary to hand-design a different network for each experiment / environment?

Minor:
* Missing space before `(` and `[`. This is repeated at least 10 times throughout the paper, e.g. lines 33 and 34.
* Line 109: repeat what ESPL stands for.
* Line 210: mixing up $t$ and $T$.


**Questions:**

1. Is Figure 1 right the exact architecture used in the experiments? Or do you hand-design a different arrangement of activations for each experiment?

2. Is it possible to scale symbolic regression to high-dimensional state spaces?

**Limitations:**

The limitations of ESPL are not explicitly addressed. I am especially curious about the possibly of scaling up to high-dimensional (e.g. image) state spaces.

---

> ### Author Rebuttal · Authors · 2023-08-08
>
> We appreciate your positive review, insightful feedback and constructive comments that help improve the quality of the paper! We are glad to answer your questions and would appreciate any further response.
>
> > **Q1.** The discovered symbolic policies in Table 1 seem to be somewhat more complex than the policies from DSP. It could be useful to additionally measure the complexity of the discovered policies for the various methods (if such measure exists).
>
> A1. Thanks for your advice.
> Because there was no well-defined measure in the previous work, we compare the length of the symbolic policies and define:
> $length=\\frac{\\sum^{i=n}_{i=1}N_o+N_c+N_v}{n}$, where n is the dimension of the action, $N_o$ is the number of operators, $N_c$ is the number of constant terms, $N$ is the number of variable terms. We give a comparison of the length of the symbolic policies in the following table.
>
> |  | Average | CartPole | MountainCar | Pendulum | InvDoublePend | InvPendSwingup | LunarLander | Hopper | BipedalWalker |
> |---|---|---|---|---|---|---|---|---|---|
> | ESPL | 12.91 | 3 | 6 | 7 | 15 | 13 | 16.5 | 24.6 | 17 |
> | DSP | 8.25 | 3 | 4 | 8 | 1 | 19 | 6.5 | 12 | 12.5 |
>
> In the benchmark environments used in the literature, in some environments ESPL produces longer symbolic policies than DSP, in others ESPL produces similar or shorter symbolic policies than DSP. In general, symbolic policies produced by ESPL are only slightly longer than DSP's, and this degree of difference has little effect for the symbolic policies (with similar interpretability in global rebuttal 2).
>
> > **Q2.** It is not very clear how much effort went into hand-picking the activation functions in each layer of the network. Is it necessary to hand-design a different network for each experiment / environment?
>
> A2. We did not hand-design a different network for each environment. We use the same symbolic network structure for all environments (as stated in lines 154-155), and use the same set of symbolic operators as the DSP for a fair comparison.
>
> > **Q3.** Is Figure 1 right the exact architecture used in the experiments? Or do you hand-design a different arrangement of activations for each experiment?
>
> A3. Yes, we use the same architecture shown in Figure 1 right for all environments. We don't need to hand-design a different arrangement of activations for each environment.
>
> > **Q4.** The limitations of ESPL are not explicitly addressed. I am especially curious about the possibility of scaling up to high-dimensional (e.g. image) state spaces.
>
> A4. ESPL is currently designed for continuous action space environments with state vector inputs, but not for environments with discrete action space or high-dimensional observation inputs. As described in Appendix G,
> for tasks with high-dimensional observation like images, the proposed ESPL and CSP can not directly generate a symbolic policy. But we can employ a neural network to extract the environmental variables and generate symbolic policy based on these environmental variables. Methods such as beta-VAE [1] or Slot Attention [2] can be used to extract environmental variables from high-dimensional observations.  We leave this in future work.
>
> [1] Higgins, Irina, et al. "beta-vae: Learning basic visual concepts with a constrained variational framework." International conference on learning representations. 2016.
> [2] Locatello, Francesco, et al. "Object-centric learning with slot attention." Advances in Neural Information Processing Systems 33 (2020): 11525-11538.
>
> >**Q5.** Is it possible to scale symbolic regression to high-dimensional state spaces
>
> Please refer to A4.
>
> Thanks aging for your detailed review. We will revise the symbols, expressions and formatting errors in the paper in revision.

---

> > ### Comment · Reviewer_iXeo · 2023-08-20
> > **Response**
> >
> > Thank you for answering my questions, I am in favor of accepting the paper.

---

### Author Rebuttal · Authors · 2023-08-08

> **Theoretical analysis of cartpole.**

The dynamic of cartpole system can be defined as:
$\\ddot{x}=\\frac{8fa+2m \\sin \\theta(4L\\dot{\\theta}^2-3g\\cos \\theta)}{8M-3m \\cos2\\theta+5m}$
$\\ddot{\\theta}= \\frac{g \\sin \\theta- ( \\cos \\theta(fa+Lm \\dot{ \\theta}^2 \\sin \\theta))/(m+M)}{L( 4/3- (m \\cos^2\\theta)/(m+M))}$

Where $f$ is the coefficient of force, $a$ represents the action, $m$ is the weight of the pole, $M$ is the weight of the cart, $L$ is the half-pole length, $\\theta$ is the angle between the pole and the vertical direction, and $x$ denotes the horizontal coordinate of the cart.

Define $X=\\begin{bmatrix} x\\\\ \\dot{x}\\\\ \\theta\\\\ \\dot{\\theta}
\\end{bmatrix}$, then the derivative of $X$: $\\dot{X}=\\begin{bmatrix} \\dot{x}\\\\ \\ddot{x}\\\\ \\dot{\\theta}\\\\ \\ddot{\\theta}
\\end{bmatrix}=\\begin{bmatrix} \\dot{x}\\\\\\frac{8fa+2m\\sin\\theta(4L\\dot{\\theta}^2-3g\\cos \\theta)}{8M-3m\\cos2\\theta+5m}\\\\\\dot{\\theta}\\\\\\frac{g \\sin \\theta- ( \\cos \\theta(fa+Lm \\dot{ \\theta}^2 \\sin \\theta))/(m+M)}{L( 4/3- (m \\cos^2\\theta)/(m+M))}
\\end{bmatrix}$
According to the Hartman-Grobman theorem, the local stability of this nonlinear system near its equilibrium point is equivalent to the linearized system near the equilibrium point. For cartpole system, the equilibrium point is $[x,\\dot{x},\\theta,\\dot{\\theta}]=[0,0,0,0]$.
If $a = 0$, the system can be linearized as:

$\\dot{X}=\\begin{bmatrix}0&1&0&0\\\\0&0&\\frac{-6gm}{8M+2m}&0\\\\0&0&0&1\\\\0&0&\\frac{g}{L(4/3-M/(m+M))}&0\\end{bmatrix}\\begin{bmatrix} x\\\\\\dot{x}\\\\\\theta\\\\\\dot{\\theta}
\\end{bmatrix}$
Calculate its eigenvalues:
$[0, 0, 3.97114593, -3.97114593]$.
Due to the presence of positive eigenvalues, according to the Hartman-Grobman theorem, the system is unstable.
If $a=17.17\\theta+1.2\\dot{\\theta}$ which is learned by ESPL,
linearize the system near the equilibrium point:
$\\dot{X}=\\begin{bmatrix}0&1&0&0\\\\0&0&\\frac{137.36f-6gm}{8M+2m}&\\frac{9.6f}{8M+2m}\\\\0&0&0&1\\\\0&0&\\frac{g-17.17f/(m+M)}{L(4/3-M/(m+M))}&\\frac{-1.2f}{L(m+M)(4/3-M/(m+M))}\\end{bmatrix}\\begin{bmatrix} x\\\\\\dot{x}\\\\\\theta\\\\\\dot{\\theta}
\\end{bmatrix}$
Calculate its eigenvalues:
$[0+0.j, 0+0.j, -26.34+6.65014286j, -26.34-6.65014286j]$.
Since all the real parts of the eigenvalues are non-positive, according to the Hartman-Grobman theorem, the system is stable.
Therefore, for the CartPole environment, the policies learned through ESPL can maintain the stability of the CartPole system.

> **Human study for interpretability.**

The assessment of policy interpretability actually requires some understanding of the environments and what each state variable means. We invited ten researchers to rate the interpretability of policies, with a maximum of 20 minutes for each policy. On a five-point scale, we told them that the interpretability could be judged based on whether they could see from the policy expression what the policies was based on to make decisions and the possible corresponding relationships between actions and states. A score of five indicated that the strategy was highly interpretable and could be designed by humans, while a score of zero indicated that it was completely uninterpretable, just like the neural network policies. We measured the average score of the interpretability of the policies. The interpretability score obtained by ESPL and DSP are:
|  | CartPole | MountainCar | Pendulum | InvDoublePend | InvPendSwingup | LunarLander | Hopper | BipedalWalker |
|---|---|---|---|---|---|---|---|---|
| ESPL | 5 | 5 | 4.5 | 4.1 | 4.3 | 3.9 | 2.6 | 3.1 |
| DSP | 5 | 5 | 4.5 | 5.0 | 4.2 | 4.0 | 3.1 | 3.2 |

Based on the results of the human-study, we found that when the symbolic expressions were relatively short, there was a consensus that the expression were highly interpretable. When the expression length became longer, the interpretability score decreased. It may be relatively difficult to
understand symbolic expressions in a short time, but it is generally believed that interpretability is much higher than black box, at least they can understand the action will be affected by which state, and the partial correlation.

> **Clarification of metaRL**

 We use the same metaRL objective, training, and testing settings as in previous works[1][2]. The context refers to the trajectories collected from the environment, and the context encoder derives a context variable based on this context. The context variable can be understood as a kind of identifier for the environment or task. During the training process, the policy maximizes rewards on the corresponding task based on the context variable. Therefore, we can anticipate that in a new task, as long as trajectories are collected and the context variable is obtained, metaRL can exhibit strong performance on that task. We also provide the pseudocode for the training process in Appendix Algorithm 2. Furthermore, the testing process of meta-RL is similar to your description: during testing, for a new task, the agent first undergoes an exploration phase to collect trajectories and update the context variable (lines 213-214). In subsequent phases, metaRL can achieve favorable results on the task based on the context variable. We will revise the description for clarity in the revision. Due to space constraints and the well-defined nature of the context-based metaRL process in the literature, we only provided a brief overview in the paper. We will add a detailed introduction to metaRL in the appendix.

[1] Rakelly, Kate, et al. "Efficient off-policy meta-reinforcement learning via probabilistic context variables." International conference on machine learning. PMLR, 2019.
[2] Sarafian, Elad, Shai Keynan, and Sarit Kraus. "Recomposing the reinforcement learning building blocks with hypernetworks." International Conference on Machine Learning. PMLR, 2021

---

### Comment · Reviewer_9VjU · 2023-08-20
**Shall we discuss？**

It seems that now we are tending to accept this draft.

---

### Decision · Program_Chairs · 2023-09-21

**Decision:**

Accept (poster)

**Comment:**

The reviews are unanimous. The merits of this paper is two-folded. First of all, it is important to explore symbolic methods for interpretability and safety guarantees. It is in general more accessible to have certificate over symbolic methods instead of neural ones. Secondly, the results look promising, and the proposed method is general. However, most of reviews have concerns (even after rebuttal and discussion) over two aspects of this paper, which we highly recommend this paper to continue exploring in the future. First of all, its applicability in more complex domains (aka, scalability issues). Secondly, is interpretability strongly correlated to symbolic methods. Or in other words, are symbolic methods always interpretable. As raised in a few reviews, the policy learned here sometimes still looks messy. Is it a sign of overfitting? We would like in the final version, this paper could consider having a more thorough and comprehensive discussion over those important issues.